# Chitosan Nanoparticle Encapsulation of Antibacterial Essential Oils

**DOI:** 10.3390/mi13081265

**Published:** 2022-08-06

**Authors:** Arvind Negi, Kavindra Kumar Kesari

**Affiliations:** 1Department of Bioproduct and Biosystems, School of Chemical Engineering, Aalto University, 02150 Espoo, Finland; 2Department of Applied Physics, School of Science, Aalto University, 02150 Espoo, Finland

**Keywords:** chitosan nanoparticles, functional chitosan materials, antibacterial, essential oils

## Abstract

Chitosan is the most suitable encapsulation polymer because of its natural abundance, biodegradability, and surface functional groups in the form of free NH_2_ groups. The presence of NH_2_ groups allows for the facile grafting of functionalized molecules onto the chitosan surface, resulting in multifunctional materialistic applications. Quaternization of chitosan’s free amino is one of the typical chemical modifications commonly achieved under acidic conditions. This quaternization improves its ionic character, making it ready for ionic–ionic surface modification. Although the cationic nature of chitosan alone exhibits antibacterial activity because of its interaction with negatively-charged bacterial membranes, the nanoscale size of chitosan further amplifies its antibiofilm activity. Additionally, the researcher used chitosan nanoparticles as polymeric materials to encapsulate antibiofilm agents (such as antibiotics and natural phytochemicals), serving as an excellent strategy to combat biofilm-based secondary infections. This paper provided a summary of available carbohydrate-based biopolymers as antibiofilm materials. Furthermore, the paper focuses on chitosan nanoparticle-based encapsulation of basil essential oil (*Ocimum basilicum*), mandarin essential oil (*Citrus reticulata*), *Carum copticum* essential oil (“Ajwain”), dill plant seed essential oil (*Anethum graveolens*), peppermint oil (*Mentha piperita*), green tea oil (*Camellia sinensis*), cardamom essential oil, clove essential oil (*Eugenia caryophyllata*), cumin seed essential oil (*Cuminum cyminum*), lemongrass essential oil (*Cymbopogon commutatus*), summer savory essential oil (*Satureja hortensis*), thyme essential oil, cinnamomum essential oil (*Cinnamomum zeylanicum*), and nettle essential oil (*Urtica dioica*). Additionally, chitosan nanoparticles are used for the encapsulation of the major essential components carvacrol and cinnamaldehyde, the encapsulation of an *oil-in-water* nanoemulsion of eucalyptus oil (*Eucalyptus globulus*), the encapsulation of a mandarin essential oil nanoemulsion, and the electrospinning nanofiber of collagen hydrolysate–chitosan with lemon balm (*Melissa officinalis*) and dill (*Anethum graveolens*) essential oil.

## 1. Introduction

The exponential growth of the human population has led to an abrupt expansion of antibiotic use and has precipitated the severity of antimicrobial resistance. The extensive use of antibiotics in the food and agriculture industry [1], and the sudden rise in patients receiving immune therapies and implants [2,3] are the leading factors of antimicrobial resistance. Additionally, the practice of poor medical policies [4] in the low-income regions of the Asia-Pacific, Africa, and South America [5] regarding the public usage of antibiotics has diverted the interest of antibiotics-manufacturing pharma industries towards conventional research areas [6], as evidenced by a comparison of drugs approved by the FDA in the last 40 years [7]. Figure 1 illustrates a comparison showing a paradigm shift as the antibiotics sector shrinks in relation to the research interest of pharma companies. Between 2015 and 2021, the FDA approved around 29% anticancer drugs, 12% neurological drugs, and 14% infectious drugs. The number of approved infectious drugs is alarmingly low based on the inclusion of various types of infectious agents (viruses, different types of bacteria, and protozoa).

## 2. Concept of Biofilms Associated with Human Health Risks

The evolution of new bacterial strains (such as multidrug-resistant (MDR) or extremely drug-resistant strains (XDR)) precipitates the threat of antimicrobial resistance to the world’s health and economy. Additionally, in-hospital patients with surgical implants have a higher rate of developing secondary infections. In most cases, these secondary infections/co-infections are linked with the dwelling microbial population (also known as microbial biofilm), which resides on the surfaces of these implants. These microbial biofilms can commonly be found in industrial or potable water piping systems. The biofilms comprise about 90% of microbial biomass of bacteria in the world [8], often producing secondary or nosocomial infections in patients with delineating diseases or histories of chronic disorders. According to CDC, an estimated 687,000 incidences and 72,000 deaths with secondary infections were recorded in the United States in 2015 [9].

To make matters worse, dwelling microbial populations in these biofilms have enhanced gene transcription to their respective planktonic counterparts (freely suspended cellular form). Therefore, a high dose of antibiotics or disinfectants is typically required to sanitize these surfaces, which leads to antimicrobial resistance among these bacteria and damages the microbial ecology. These biofilms are an assemblage of microbial cells that are irreversibly associated with a surface and enclosed in a matrix primarily made up of polysaccharides [10]. The composition of biofilms is heterogeneous, containing microcolonies of various bacteria strains encased in a polysaccharide-based matrix, also called an extracellular polymeric substance (abbreviated as EPS) matrix. However, it is a continuous monolayer surface deposit; therefore, it is a random order of bacteria (a schematic representation of biofilm-based human infections is shown in Figure 2).

Typical bacteria produce secondary infections from these biofilms, mainly containing Gram-positive bacteria (such as *Enterococcus faecalis*, *Staphylococcus aureus*, *Staphylococcus epidermidis*, and *Streptococcus mutants*) or/and Gram-negative bacteria (such as *Escherichia coli*, *Klebsiella pneumoniae*, *Proteus mirabilis*, and *Pseudomonas aeruginosa*). For example, scanning electron microscopy images of *Staphylococcus aureus* biofilm formation in vivo on a titanium orthopedic implant are shown in Figure 3, where the left-hand side shows an irregular pattern of biofilm formation (marked by yellow arrows in Figure 3) on the concave surface of the implant, while the right-hand side shows the appearance of several cocci (shown in round-shaped yellow color) interspersed with matrix material.

### Composition of Biofilms: Syntropy of Bacteria

Biofilms comprise extracellular polymeric substances (abbreviated as EPS), mainly polysaccharides. In Gram-negative bacteria, some polysaccharides could be neutral or polyanionic. However, the polyanionic nature of these polysaccharides is characterized primarily by uronic acids (such as D-glucuronic, D-galacturonic, and mannuronic acids) or ketal-linked pyruvate and, therefore, imparts a negative charge over the microbial biofilm surfaces [13]. This negative charge over the microbial biofilm surface helps in the association of these biofilm layers with divalent metal cations such as Ca^2+^ and Mg^2+^, which further enhances cross-linking among these polymers and the development of biofilm [14]. On the contrary, in some Gram-positive bacteria, such as the staphylococci, the chemical composition of EPS could be different due to their cationic nature. Apart from polysaccharides, biofilm contains lipids, proteins, nucleic acids, and humic substances. EPS could be hydrophobic or amphiphilic. However, the presence of a large number of hydroxyl groups of polysaccharides in EPS incorporates water molecules via hydrogen bonding; therefore, the composition and structure of these polysaccharides play an essential role in determining their primary conformation and physicochemical functions of the biofilm [10], e.g., bacterial EPS, which contains a backbone of 1,3- or 1,4-β-linked hexose residues, characterizes low solubility.

Interestingly, EPS is not uniformly distributed and is also attributed by the age of biofilm and concentration of metal ions and macromolecules, and even its biosynthesis can severely be limited or enhanced by levels of carbon, nitrogen, potassium, and phosphate [13]. The dynamicity of composition in biofilm architecture is continuously affected by internal and external processes. As microcolonies of bacteria live in syntropy, they continuously exchange their genetic material or antibiotic-resistant gene, quorum sensing, and cycling of nutrients.

In conclusion, bacteria of biofilms are difficult to target through conventional antibiotics. Secondly, secondary infections are prominently influenced by these bacteria of biofilms; therefore, much interest has been drawn towards the exploration of polymeric materials that can be used to make commercial biomedical products. While several examples of polymeric materials have been cited in the literature, their lower availability, higher cost, and biodegradability has restricted their direct use to becoming a preferred choice of interest.

## 3. Natural Polymeric Carbohydrate-Based Antibiofilm Materials

Natural biopolymers, especially carbohydrates, show excellent material properties, such as mechanical strength, plasticity, and biodegradability. In addition, the anionic charges of hydrophilic polysaccharides (such as sulfate polysaccharides, hyaluronic acid, etc.) allow ionic attraction with metal ions or organic salts, and thereby they achieve antibacterial material properties.

### 3.1. Sulfated Polysaccharides

A sulfated polysaccharide (fucoidan) isolated from marine algae (*Fucus vesiculosus*) showed activity against biofilm-forming bacteria [15,16], as shown in Figure 4A. The minimum inhibitory concentrations against food-borne bacteria (*Listeria monocytogenes* = 250 μg mL^−1^; *Staphylococcus aureus* = 500 μg mL^−1^) and dental plaque bacteria (*Enterococcus faecalis =* 1000 μg mL^−1^, *Streptococcus mutans* = 125 μg mL^−1^; *Streptococcus oralis* = 500 μg mL^−1^; *Streptococcus sobrinus* = 250 μg mL^−1^) have been recorded [15]. Figure 4B shows the structure of dermatan sulfate (glycosaminoglycan), and its co-immobilization with chitosan on polyethylene terephthalate surfaces prevents *Staphylococcus epidermidis* from forming a biofilm.

In another study, Lentina, a mushroom polysaccharide, derivatized with the help of dimethyl sulfoxide into sulfate form [17], is shown in Figure 4C. This chemical transformation enhanced the water-solubility of the sulfated form as anionic polymers, which made it possible to combine with polycationic chitosan deposited onto the surfaces of polyurethane (PU) via the layer-by-layer (LbL) assembly technique [17]. These polymeric coatings showed a significant inhibition of the growth of *Pseudomonas aeruginosa* and reduced fibrinogen adsorption and platelet adhesion [17]. In other study, an extract containing sulfated polysaccharides (SPs) from green algae (*Chlamydomonas reinhardtii*) was studied for antibacterial and antibiofilm activity at 0.5 mg mL^−1^. It exhibited 34.52, 48.6, 66.1, and 55.6% reduced colony-forming units (CFU) against *Bacillus subtilis*, *Streptococcus*, *Neisseria mucosa*, and *Escherichia coli*, respectively [18].

### 3.2. Hyaluronic Acid

A grafted copolymer derivative (HA-EDA-BMP-MANa) of hyaluronic acid (HA) with ethylamine (EDA), and a methyl propionic acid (BMP) polymethacrylate structure (MANa, as shown in Figure 5) was successfully reported using atom transfer radical polymerization (ATRP chemistry) [19,20], as shown in Figure 5. The presence of ionic carboxylic and amino groups in the prepared copolymer (HA-EDA-BMP-MANa) directed its formulation (at three different pH values, namely, 5, 6, and 7) into a hydrogel at a concentration of 10% *w*/*v*, with or without vancomycin (2% *w*/*v*). In addition, a sustain pH-dependent in vitro vancomycin release over 48 h was observed, exhibiting biomaterial improved properties against *Staphylococcus aureus* adhesion on titanium disks compared to the other reported unmodified hyaluronic acid hydrogel [21].

### 3.3. Pectins

Pectins are polysaccharides consisting primarily of esterified D-galacturonic acid that resides in an α-(1–4) chain, as shown in Figure 6. Alkaline processes hydrolyze ester groups, exposing more carboxylic groups for ionic interaction with metals (silver nanoparticles [22,23]) or organic salts (benzalkonium chloride [24], dodecyl trimethylammonium chloride [25]) and imparts antibiofilm activity.

### 3.4. Dextran

Dextran is a long-chain glucose homopolysaccharide with primarily α-1,6-glucopyranosidic linkages (shown in Figure 6) and is produced from sucrose by *Leuconostoc mesenteroides* and related bacteria, and from dextrin by other bacteria. Its candidature as an implant material, dextran-70 (mol. mass ≈ 70 000), can be retained in the intravascular spaces and contribute to the colloid plasma oncotic pressure. However, in high concentrations, dextran-70 inhibits the aggregation of platelets and enables fibrinolysis. In addition, dextran–chitosan gel showed antibiofilm activity [26]; Li et al. reported an acid-induced self-catalyzing material (dextran-coated copper peroxide nanoaggregates (DCPNAs) for antibiofilm [27]; Hoque et al. reported sustained-release antibacterial dextran hydrogels to eradicate the microbial biofilms [28]; Naha et al. reported dextran-coated iron oxide nanoparticles for pH-based biofilm disruption [29].

### 3.5. κ-Carrageenan

Structurally, κ-carrageenan is a linear sulfated polysaccharide (shown in Figure 7) commonly found in red seaweed. Green microwave synthesis was used to form κ-carrageenan–silver nanoparticle (CRG–Ag) nanocomposites of size 50 ± 10 nm, and they were found to be active against *Staphylococcus aureus*- and *Pseudomonas aeruginosa*-mediated biofilms [30]. Additionally, CRG–Ag nanoparticles encapsulated in potassium chloride cross-linked hydrogel displayed reasonable thermal stability and antimicrobial activity [30].

## 4. Chitosan as Polymeric Material Surface

Chitosan is the second most abundant polysaccharide in nature. Most of its natural proportion is found in the form of the exoskeletal coat of arthropods [31,32]. Its remarkable physicochemical properties, such as biocompatibility, biodegradability, and low toxicity, make chitosan a suitable material for various biomedical uses [32,33]. It is an amino polysaccharide linear polycation biopolymer (as shown in Figure 8), randomly distributed β-(1→4)-linked D-glucosamine (deacetylated unit) and N-acetyl-D-glucosamine (acetylated unit) with a high charge density, reactive hydroxyl and amino groups, along with extensive hydrogen bonding, which enhances its physical stability and processability [33,34]. However, despite its characteristics, it exhibits poor solubility due to extensive hydrogen bonds and acetamido groups in the crystalline state. The presence of polycationic charges on chitosan imparts its antimicrobial properties. These positively charged amino groups of glucosamine of chitosan interact with negatively charged membrane constituents of microbes, inducing leakage of intracellular materials of the cell and disrupting the membrane function, as found in Gram-negative bacteria (including *Escherichia coli* and *Salmonella typhimurium*), Gram-positive bacteria (*Staphylococcus aureus)* [35], and yeast (*Saccharomyces cerevisiae*) [36] [33,37].

### Different Forms of Chitosan and its Quaternization as Antibiofilm Activity

Various forms of chitosan have shown antibiofilm activity [38]. Factors such as the molecular weight (high molecular weight HMW, medium molecular weight MMW, low molecular weight LMW), degree of deacetylation/acetylation, and derivatization affect the antibiofilm activity of chitosan [38]. Both versions of chitosan—HMW (624 KDa) with a degree of deacetylation >75%, and LMW (107 KDa) with a degree of deacetylation of 75–85%—inhibit the adhesion and maturity of *S. mutants* [39] and prevent the growth of planktonic cells and adhesion of vancomycin-resistant *Staphylococcus aureus* and *Enterococcus faecalis* [40]. Furthermore, LMW (107 KDa, with a degree of deacetylation of 75–85%) chitosan prevented the growth, adhesion, and biofilm formation of methicillin-resistant *Staphylococcus aureus* (MRSA), methicillin-susceptible *Staphylococcus aureus* (MSSA), and methicillin-resistant *Staphylococcus epidermidis* (MRSE). These studies verify the intrinsic antibiofilm potency of chitosan [38].

To enhance the antibiofilm activity of chitosan, researchers attempted various derivatizations: (a) carboxymethyl chitosan (30 KDa) with a degree of deacetylation (90%) inhibited the broad spectrum antibiofilm activity of *Lactobacillus gasseri*, *Streptococcus salivarius*, *Rothia dentocariosa*, and *Staphylococcus epidermidis* [41]; 2-methylaziridine-modified chitooligosaccharide changed the fluidity of biofilm and the bacterial cell membrane of *Pseudomonas aeruginosa* by releasing nitric oxide [42]. Among derivations, quaternization is a common strategy as it provides an ionic charge that can be used for adsorption of the oppositely charged ions/functional molecules. For example, hydroxypropyl-trimethyl ammonium chloride chitosan (HACC), with a degree of deacetylation of 91.83%, prevented biofilm growth of *Staphylococcus aureus* and *Staphylococcus epidermidis* by targeting the extracellular polysaccharides-encoded gene (*icaA*) expression [43]; (b) N,N,N-trimethylchitosans showed broad-spectrum antibiofilm activity against Gram-positive (*Staphylococcus epidermidis*) and Gram-negative bacteria (*Escherichia coli*) [44]; (c) Quaternary ammoniumyl chitosan (degree of deacetylation = 96%) derivatives prevented the biofilms of *Staphylococcus aureus* [45] and other examples, as reported in the following reports [46,47].

## 5. Encapsulation of Phytochemicals/Essential Oils

Phytochemicals always have been a source of medicinally active compounds [48], such as anticancer indole alkaloids and antimalarial cassiarins [49]. However, phytochemicals lack essential structural features, making their physicochemical profile challenging in terms of compliance with the “Lipinski rule of 5” (which evaluates the druggability) [50]. In addition, most essential oils have typical hydrophobicity, restricting their direct biomedical use, and leading to the development of various formulations in recent years. One such approach is encapsulation at the microscale [51] and nanoscale [52]. As is typical, polymeric material is used to encapsulate these phytochemicals, and successful integration proportionally depends on particle size; therefore, nanoscale formulations were preferred over microscale formulations.

Researchers commonly cited the proof of concept that the extracted essential oils have activity against the common bacteria of the biofilm [53,54]. Furthermore, studies revealed the effectiveness of nanoemulsions in preventing biofilm formation, loosening the cells from the initial biofilm, and amplifying the drugs’ antibacterial activities [55,56,57]. Furthermore, the nanoscale size of nanoparticles is small enough to penetrate the extracellular polymeric substances (EPSs) of biofilm and prevent biofilm formation [58].

Various methods used for chitosan nanoparticles, which can be categorized based on production method and matric composition, are shown in Table 1. The ionic gelation method is commonly practiced for the nanoencapsulation of functional compounds (essential oils, food, cosmetics, drugs, etc.). Chitosan has become the most obvious choice of many researchers because of its biodegradable nature, easy accessibility, and availability of surface functional groups (amino groups). Because of these amino groups, chitosan can be cross-linked easily to other functional materials. Key choices of crosslinkers are TPP (sodium tripolyphosphate) and HMP (hydroxymethyl melamine prepolymer). Various factors come into play to achieve effective chitosan-based nanoencapsulation, such as pH, the solvent used, mixing sequence, concentration, and molar ratios. However, the most critical aspect for succeeding in chitosan-based nanoencapsulation is measured by encapsulation efficiency (EE) and loading capacity (LC). By definition, encapsulation efficiency is measured in percentage, which indicates how much fractional amount of functional compound (essential oil, drugs, edible compounds, etc.) is entrapped (encapsulated) into the nanoparticles (called nanoencapsulation) or micelles, whereas loading capacity (LC) reflects a fraction of the amount of encapsulated functional material over the total weight of nanomaterial support used.

### 5.1. Ocimum basilicum L. Essential Oil

Yu et al. from Dalian Minzu University (Dalian, China) encapsulated the *Ocimum basilicum* L. essential oil (BEO) into chitosan nanoparticles (CSNPs) by emulsion and ionic gelation [93]. To their rationality, the use of chitosan serves as a carrier and improves the formulation’s biological properties (biocompatibility, safety, and degradability). Two steps (emulsification and ionic gelation) were used to prepare the BEO-loaded chitosan nanoparticles [94]. The choice of *Escherichia coli* and *Staphylococcus aureus* to measure the antibiofilm activity was evident based on their high biofilm-forming tendency and higher proportional population in biofilm microcolonies. The particle size, polydispersity index (PDI), and zeta potential of BEO-loaded CSNPs were accessed through the dynamic light scattering (DLS) method.

The GC–MS data showed seventeen compounds that accounted for 95.5% of the basil essential oil, where eugenol (48.32%) and caryophyllene (26.26%) were found to be prominent ones, and the remaining ones were less than 6%. The CSNPs with no BEO had a particle size of 198.7 nm. The authors made three combinations of BEO with CSNPs, where they chose a ratio (CSNP:BEO = 1:0.5; zeta potential = 30.7 ± 0.47 mV) based on the conditional requirement of particle surface charge (zeta potential above 30 mV). A particle surface charge over CSNPs is vital to achieving specific colloidal solution characteristics (aggregation, dispersion, and flocculation) and can directly affect the bioavailability of the encapsulated essential oils [93]. Furthermore, the nanoscale characteristic was compared using scanning electron microscopy (SEM) images, as shown in Figure 9.

Ultraviolet–visible (UV–Vis) spectroscopy was used to estimate the encapsulation efficiency and loading capacity. The authors chose the CSNPs loaded with BEO (1:0.5), as it exhibited excellent encapsulation efficiency (75.13 ± 0.09%) and loading capacity (18.63 ± 0.02%).
$$\mathrm{EE}\;\left(\%\right)=\frac{\mathrm{Total}\;\mathrm{amount}\;\mathrm{of}\;\mathrm{Loaded}\;\mathrm{BEO}.}{\mathrm{Initial}\;\mathrm{Amount}\;\mathrm{of}\;\mathrm{Loaded}\;\mathrm{BEO}.}\times 100$$
$$\mathrm{LC}\;\left(\%\right)=\frac{\mathrm{Total}\;\mathrm{amount}\;\mathrm{of}\;\mathrm{Loaded}\;\mathrm{BEO}.}{\mathrm{Weight}\;\mathrm{of}\;\mathrm{Freeze}-\mathrm{Dried}\;\mathrm{nanoparticles}}\times 100$$

*E. coli* and *S. aureus* were used to evaluate the antibacterial activity of chitosan powder, unloaded CSNPs, and BEO-loaded CSNPs. In comparison, unloaded CSNPs showed improvement (*E. coli* = 46.67 ± 4.71%; *S.*
*aureus* = 25.76 ± 6.88%) with BEO-loaded CSNPs (*E. coli* = 78.33 ± 12.96%; *S. aureus* = 80.81 ± 19.99%). To understand the antibacterial mechanism of BEO-loaded CSNPs, a DNA leakage study on *E. coli* and *S. aureus* was performed. The change in DNA leakage posttreatment with BEO–CSNPs for 8 h increased by 60.76% against *E. coli* and 50.88% against *S. aureus* [93]. While some sugars (such as iminosugars) are mechanistically found as metabolic inhibitors [95].

### 5.2. Mandarin Essential Oil 

Wu et al., from Shaanxi Normal University (Shaanxi, China), encapsulated mandarin essential oil (MEO) (*Citrus reticulata*) with CSNPs to improve antibacterial properties and prolong pork preservation [96]. Various loading capacities of MEO were used with CSNPs with the following CSNP:MEO ratios: 1:0, represented as CSs-C; 1:0.2, represented as CSs-L; 1:0.5, represented as CSs-M; and 1:1, represented as CSs-H. The encapsulation efficiency (EE) for MEO-CSNPs was found to be 67.32–82.35%, the mean particle size was 131.3 nm–161.9 nm, and the zeta potential was 30 mV.
$$EE\;\left(\%\right)=\frac{\mathrm{Total}\;\mathrm{amount}\;\mathrm{of}\;\mathrm{Loaded}\;\mathrm{MEO}}{\mathrm{Initial}\;\mathrm{Amount}\;\mathrm{of}\;\mathrm{Loaded}\;\mathrm{MEO}}\times 100$$

The mean particle size is correlated with essential oil type, (TPP) concentration, and wall material [97]. Zeta potentials evaluate the aggregation/dispersion between particles. As in all cases, zeta potential exceeded 30 mV, which suggested that a sufficient electrostatic repulsion among the droplets stabilized them in the MEO-loaded CSNPs [98]. However, zeta potential and particle size directly affect the antibacterial activity of nanoencapsulation and influence its interaction with multiple molecular sites [99]. Hence, these results demonstrate MEO-loaded CSNPs’ stability in the emulsion state, although the addition of MEO altered the nanoscale size of CSNPs.

All FTIR spectra show regions affiliated with the functional groups 664 cm^−1^ (pyranose ring of chitosan), 1079 cm^−1^ (C–N stretching), 1379 cm^−1^ (C–O–H, H–C–H), and 2875 cm^−1^ (C–H stretching), as shown in Figure 10A. The appearance of the 1230 cm^−1^ (–P=O stretching) peak in the spectra indicates the TPP cross-linking in CSNP:MEO ratios, whereas XRD analysis exhibits two peaks (at 20° and 29°) representing the crystallinity of chitosan; however, the peak intensity faded in samples loaded with MEO denoting the complex structural changes in these ratios. The 1:1 ratio of chitosan:MEO shows the most potent antibacterial activity against *Staphylococcus aureus* and *Escherichia coli*. Furthermore, the authors evaluated the effect of such nanoencapsulation on bacterial cell morphology. It was found out that increases in MEO loading with CSNPs damage the bacterial cell morphology (*Staphylococcus aureus* and *Escherichia coli*) and exhibit irregular, deformed, and incomplete structures. However, the severity of damage was noticed with *Staphylococcus aureus* compared to *Escherichia coli*, reflecting the differences in their cellular makeup. *Escherichia coli* is Gram-negative, whereas *Staphylococcus aureus* is Gram-positive, reflecting physicochemical property changes because of the macromolecular change of the distinctive membrane. The authors further evaluated the effect of MEO-loaded CSNPs on biofilm formation. In this test, the initial stage of adhesion of bacteria was evaluated, as it is difficult to remove the bacteria once they achieve this stage. Increased loadings of MEO:CSNPs showed potent activity against the initial adhesion stage. Environmental scanning electron microscopy (ESEM) analysis was used to compare the untreated and treated groups with MEO:CSNPs. In this analysis, the untreated group showed a typical mature biofilm, where aggregated bacteria were found in the polysaccharide structure of biofilm (higher thickness), while treated groups were found with lower thickness, showing less aggregation of bacteria. The plate counting method was further tested for MEO:CSNPs for their activity against the mature biofilm. An efficient destructive percentage of biofilm was observed for a 1:1 ratio of MEO:CSNPs and a notable reduction in the cell population.

In another study, a collaboration with the Competence Center on Agro-Food Productions, and Department of Industrial Engineering, University of Salerno (Italy) and the Canadian Irradiation Center, INRS—Institut Armand-Frappier, Institute of Nutraceutical and Functional Foods Québec, Canada modified chitosan containing a 0.05% mandarin essential oil nanoemulsion [100]. The antibacterial testing of samples was performed with γ-irradiation, UV-C, and ozone-treated water treatments. The combined coating and γ-irradiation showed a synergistic effect on microbial growth (3.3 log CFU/g), while a 3 log CFU/g reduction of the initial *Listeria innocua* population was observed when combined with UV-C irradiation. However, no antimicrobial effect of the combination with ozone-treated water was observed [100].

### 5.3. Carum copticum Essential Oil 

Esmaeili and Asgari from Islamic Azad University (Tehran, Iran) applied the emulsion ionic gelation method on CSNPs to encapsulate *Carum copticum* essential oil (CEO) [101]. To explore the encapsulation efficiency (EE) and loading capacity (LC), various concentrations of TPP and HMP as cross-linkers were used with chitosan. Based on EE and LC values, the authors chose a sample with a 1:1 mass ratio of chitosan to CEO and a TPP concentration of 0.5% (*w*/*v*).

FTIR was used to characterize chitosan–TPP nanoparticles (Figure 11B), as shown in Table 2. The peak of amide-II (single bond NH_2_ bending) shifted from 1590 to 1535 cm^−1^, and new peaks appeared around 1100–1290 cm^−1^ (P–O and P=O) showing the presence of phosphorus of TPP from CEO-loaded chitosan–TTP nanoparticles. Secondly, a pronounced sharpening of the C–H stretching region in FTIR further support CEO integration in chitosan nanoparticles. These changes in FTIR spectra assure the CEO encapsulation into the chitosan nanoparticles.

Thermal analysis by differential scanning calorimetry (DSC) was used to study the thermal behavior of pure CEO, chitosan nanoparticles, and CEO-loaded chitosan nanoparticles. As anticipated from the previous literature [103], the thermogram of chitosan showed an endothermic peak (at 75 °C) related to the loss of adsorbed water (associated with hydrophilic groups of polymer), along with an exothermic peak (at 311 °C) reflecting its molecular degradation (dehydration of the anhydro-glycosidic ring, depolymerization and chemical decomposition of monosaccharide units). The thermogram of CEO exhibited two endothermic peaks (in the region of 35–128 °C) and at 160 °C, which can be correlated to its evaporation and the chemical degradation of low boiling-point components (Figure 11C(a)) [104,105]. Unlike with chitosan, distinctive thermal behavior was noticed in CEO-loaded CSNPs, where exothermic peaks of 145 and 219 °C of CSNP were shifted to 150 and 226 °C (Figure 11C(b,c)). Further conclusions can be drawn from this thermal behavior: (a) the recorded change reflected the change in molecular assembly CEO-loaded CSNPs, and (b) overall improved the thermal stability of these materials. These changes were also in parallel with the previously reported literature [105,106].

The SEM measurements of CSNPs and CEO-loaded CSNPs showed spherical shaped particles with an average diameter of 30–80 nm. However, a significant difference in the mean particle size and size distribution of CSNPs and CEO-loaded CSNPs by the DLS technique was found (mean diameter of CSNPs = 954 nm; average diameter of CEO-loaded CSNPs = 236.0–721.0 nm). This profound difference in the diameter indicated the method sensitivity, as the distinctive nature of SEM and DLS measurements in an aqueous solution could lead to swelling and aggregation of the CSNPs during dispersion in water, while a reduction in swelling and/or aggregation of CEO-loaded CSNPs in an aqueous solution compared to the CSNPs in an aqueous solution might be due to the hydrophobicity of CEO molecules encapsulated inside/on the nanoparticles. Similar observations were also reported in the literature [107].

The in vitro release study of CEO from CEO-loaded CSNPs was evaluated with varying pH using buffers of pH 3 and 5 (acetate buffer), pH 7.4 (phosphate buffer), and pH 10 (phosphate buffer) for 4 days. The in vitro release studies of CEO were conducted from the prepared sample with a mass ratio of chitosan to CEO of 1:1 and a TPP concentration of 0.5% (*w*/*v*). Generally, the release of encapsulated compound uses some or all of the following mechanism: diffusion, desorption, disintegration, and surface erosion [94]. However, diffusion followed by polymer matrix degradation is commonly observed in CSNP-based encapsulation [94,102]. A biphasic process where an initial burst (for 5 h) was followed by a slow declining release at all pH values was recorded until 24 h. The observed initial burst could be related to the facile release of CEO molecules either loosely bound or superficially encapsulated [94]. After 24 h, a steady release was observed in all the pH cases until 72 h. After 72 h, a significant decrease was noted in release levels, as a plateau stage was achieved. As in previous reports [108] of pH affecting the in vitro release of the encapsulated materials or compounds from the nanoparticles, the authors noticed that the CEO released in lower pH (3 and 5) buffers was significantly higher (*p* < 0.05) than that in the saline (pH 7.4) and basic buffer (pH 10). The observation that having a higher diffusion rate of CEO in an acidic medium could be correlated with the strength of ionic repulsions of protonated NH_2_ groups of chitosan with each other contributed to the partial dissolution and swelling of CSNPs [106,109]. Furthermore, as the acidity of the solution increases (pH 3 buffer), a proportional rise in the swelling will also be observed; therefore, a greater CEO release was observed in pH = 3 buffer than in pH = 5 buffer. Such observations were also reported with an in vitro release study of curcumin-loaded dextran sulphate–chitosan nanoparticles systems as well [109]. As anticipated, the swelling property of chitosan decreased with an increase in alkalinity of the solution, and a lower released amount of CEO was expected.

Contrary to the previous statement, the in vitro release of CEO found at pH = 10 buffer was found to be significantly higher (*p* < 0.05) than that at pH = 7.4 (saline buffer), reasonably due to the deprotonation of NH_2_ groups of chitosan and therefore a decline in strength of ionic repulsion [107]. Comparatively, more release of essential oil (CEO) was observed in acidic buffers than in alkaline buffers, supporting chitosan as a suitable material scaffold to control the release of the essential oils. In conclusion, the in vitro release demonstrates the role of chitosan’s physicochemical parameters, which affect its material properties as a polymer.

The antibacterial study of CSNPs and CEO-encapsulated CSNPs was studied by the agar disk diffusion method on *Staphylococcus aureus*, *Staphylococcus epidermidis*, *Bacillus cereus*, *Escherichia coli*, *Salmonella typhimurium*, and *Proteus vulgaris*, as summarized in Table 3. The opted controls (phosphate buffer saline and dimethyl sulphoxide, negative control) showed no antibacterial activity. The non-encapsulated CEO showed antibacterial activity against all the strains, as suggested in the previous study [110] (as shown in Table 3). The antibacterial activity of CEO was due to the presence of critical essential oils (thymol, γ-terpinene, and ρ-cymene) [111]. These essential oils cross the bacterial membrane and alter the cytoplasm’s pH and equilibrium of ionic concentration, leading to their antibacterial activity [112]. Compared with non-encapsulated CEO or chitosan nanoparticles (CSNPs), CEO-loaded CSNPs exhibited improved antibacterial activity, further justifying the suitability of chitosan-based material support for essential oil nanoencapsulation. Furthermore, these observations suggested a synergistic antibacterial mechanism, where positively charged pronated NH2 terminals of chitosan could easily form complexations with the negatively charged components of bacterial membrane, which leads to the swelling of the CSNP encapsulation and release of essential oils that further damage the bacterial cell [113].

### 5.4. Nanoencapsulation of Cinnamaldehyde 

Subhaswaraj et al. reported the encapsulation of cinnamaldehyde (CA) into CSNPs using the ionic gelation method [114], as shown in Figure 12. Dynamic light scattering (DLS) and transmission electron microscopic (TEM) analysis confirmed the synthesis of CA-loaded CSNPs (mean diameter = 208.12 nm). The encapsulation efficiency was estimated to be 65.04 ± 3.14%. A further in vitro release study confirmed the slow and sustained release of cinnamaldehyde. CA-loaded CSNPs showed significant anti-quorum-sensing activity by down-regulating the quorum-sensing regulated virulence factors and associated biofilm formation, as evidenced from microscopic observation. The authors tested against *Pseudomonas aeruginosa* PAO1 (minimum inhibitory concentration = 1000 µg/mL). To understand the mechanism, the authors performed a pyocyanin inhibition assay to access the production of pyocyanin, as it is an important virulence factor in these bacteria. The authors found that pyocyanin production is significantly inhibited (93.24%) with a sub-MIC concentration. CA-loaded CSNPs also considerably limited the motility of *Pseudomonas aeruginosa* [114]. The present results demonstrated an application of CA-loaded CSNPs as potential anti-quorum sensing agents compared to native cinnamaldehyde and suggested new avenues for developing novel anti-infective agents in the post-antibiotic era.

The encapsulation efficiency of CA-loaded CSNPs was found to be 65.04 ± 2.14%. However, a release study showed that 29% of cinnamaldehyde was released within 12 h, followed by 60% in the next 12 h. This release study showed a slow, sustainable release of cinnamaldehyde from CSNPs.
$$\mathrm{EE}\;\left(\%\right)=\frac{\left(\mathrm{Total}\;\mathrm{cinnamaldehyde}\;\mathrm{Loaded}-\mathrm{Nonencapsulated}\;\mathrm{Cinnamaldehyde}\right)}{\mathrm{Total}\;\mathrm{cinnamaldehyde}\;\mathrm{Loaded}}\times 100$$
$$\mathrm{Release}\;\left(\%\right)=\frac{\mathrm{Released}\;\mathrm{Cinnamaldehyde}\;\mathrm{from}\;\mathrm{CSNPs}}{\mathrm{Total}\;\mathrm{Amount}\;\mathrm{of}\;\mathrm{Cinnamaldehyde}\;\mathrm{in}\;\mathrm{CSNPs}}\times 100$$

### 5.5. Nanoencapsulation of Anethum graveolens

Researchers from the Centre of Advanced Studies in Botany (Institute of Science, Banaras Hindu University, Varanasi, India) nanoencapsulated *Anethum graveolens* seed essential oil (AGEO) within a chitosan biopolymer [115]. In addition, they studied the physicochemical properties of nanoencapsulated AGEO by SEM, XRD, and FT-IR.

The encapsulation efficiency (EE), loading capacity (LC), nanoparticle yield (NY: determined only at maximum loading of chitosan to AGEO in a ratio of 1:0.8% *w*/*v*), and release studies (cumulative release%) were estimated as per the following equations [115]:$$\mathrm{EE}\;\left(\%\right)=\frac{\mathrm{Total}\;\mathrm{amount}\;\mathrm{of}\;\mathrm{Loaded}\;\mathrm{AGEO}}{\mathrm{Weight}\;\mathrm{of}\;\mathrm{Nanoemulsion}}\times 100$$
$$\mathrm{LC}\;\left(\%\right)=\frac{\mathrm{Total}\;\mathrm{amount}\;\mathrm{of}\;\mathrm{AGEO}}{\mathrm{Intial}\;\mathrm{amount}\;\mathrm{of}\;\mathrm{AGEO}}\times 100$$
$$\mathrm{NY}.\;\left(\%\right)=\frac{\mathrm{Amount}\;\mathrm{of}\;\mathrm{freeze}\;\mathrm{dried}\;\mathrm{Nanoparticle}}{\mathrm{Sum}\;\mathrm{Total}\;\mathrm{of}\;\mathrm{All}\;\mathrm{the}\;\mathrm{individual}\;\mathrm{components}}\times 100$$
$$\mathrm{Cumulative}\;\mathrm{Release}\;\left(\%\right)=\frac{\mathrm{Cumulative}\;\mathrm{amount}\;\mathrm{of}\;\mathrm{AGEO}\;\mathrm{released}\;\mathrm{at}\;\mathrm{each}\;\mathrm{time}}{\mathrm{Initial}\;\mathrm{amount}\;\mathrm{of}\;\mathrm{AGEO}\;\mathrm{loaded}\;\mathrm{in}\;\mathrm{sample}}\times 100$$

The researchers evaluated the AGEO:chitosan nanoencapsulation for in vitro antifungal efficacy of AGEO against an aflatoxigenic strain of *Aspergillus flavus* (AF LHL R14 strain) and other food molds (*Aspergillus niger*, *Aspergillus candidus*, *Aspergillus sydowii*, *Aspergillus fumigatus*, *Arachis repens*, *Aspergillus luchuensis*, *Fusarium poae*, *Fusarium oxysporum*, *Cladosporium herbarum*, *Curvularia lunata, Alternaria alternata*, *Aspergillus humicola*, and *Mycelia sterilia*) [115].

Later, a collaborative work of Romanian researchers blended collagen hydrolysates (extracted from bovine tendons (HCB) and rabbit skin (HCR)) with chitosan (CS) by the coaxial electrospinning technique into nanofibers for encapsulating the AGEO. They evaluate the loading efficiency (LE%) using the following equation:$$\mathrm{LE}\;\left(\%\right)=\frac{\mathrm{EO}\;\mathrm{measured}\;\mathrm{amount}\;}{\mathrm{EO}\;\mathrm{theoretical}\;\mathrm{amount}}\times 100$$
where the theoretical amount of EO was 23%.

AGEO showed a narrower range of antimicrobial activity (active against *Staphylococcus*
*aureus* and *Candida glabrata*). However, nanoencapsulation of AGEO with HCB-CS (HCB-CS/AGEO) and HCR-CS (HCR-CS/AGEO) showed broad-spectrum antimicrobial activity (fungal strain: *Aspergillus brasiliensis* ATCC 9642; yeast strain: *Candida albicans* ATCC 10231; bacteria strains: *Staphylococcus aureus* ATCC 25923, *Escherichia coli* ATCC 25922, *Enterococcus faecalis* ATCC 29212, and *Salmonella typhimurium* ATCC 14028), as shown in Table 4. Furthermore, a drastic antimicrobial spectrum difference was observed with unloaded HCB-CS and HCR-CS, as shown in Table 4.

### 5.6. Nanoencapsulation of Melissa officinalis L.

The Romanian research team extended the scope of encapsulating polymeric supports (HCB-CS or HCR-CS) to lemon balm essential oil (*Melissa officinalis* L.; MOEO) to form nanofibers. The FTIR studies provided more insights into the chemical composition, as shown in Figure 13. In this figure, researchers compared the various possible versions of nanoencapsulation with/without loading of AGEO or MOEO.

Based on observations from Figure 13, the following conclusions can be drawn: (a) the amide-I functionality of chitosan (stretching vibrations of C=O groups, ν = 1639 cm^−1^) was shifted for HCB-CS (ν = 1663 cm^−1^) and HCR-CS (ν = 1635 cm^−1^); (b) amide-II (ν = 1545 cm^−1^) associated with the secondary structure in chitosan was not found in collagen–chitosan mixtures and encapsulated essential oil samples, providing evidence of participation of chitosan’s –NH_2_ and –OH groups in chemical reactions [117]; and (c) the appearance of a peak (ν = 1635 cm^−1^) in encapsulated samples suggested the presence of essential oils (AGEO and MOEO) within the electrospun collagen–chitosan nanofiber complex.

Antimicrobial testing showed a broader spectrum for encapsulated samples than unloaded essential oils (L, D&L vs. HCB-CS/L, HCB-CS/D&L, HCR-CS/L, HCR-CS/D&L), as shown in Table 5.

The in vivo studies on nanofiber samples were performed on 3-month-old white Swiss adult mice (25–30 g in weight with uniform sex distribution). Animals were supervised for a week and anesthetized using an intraperitoneal route with ketamine (50 mg/kg) and xylazine (10 mg/kg). Later, the dorsal on the left side was shaved, and a superficial incision (1 cm parallel to backbone) was made. The nanofiber textile material, as shown in Figure 14 (size: 1 × 0.5 cm), was fixed on the incision area, while a dry sterile patch was used for the control group animals.

After animals were fed and watered, personal hygiene was maintained until the 7th day. On the 7th day, patches were removed, and the incision area was evaluated under a microscope. Both animal groups (treated and control) did not show any appearance of inflammation, and the incision area was scarred. Furthermore, no hematology differences (%) were recorded for the treated group compared to the control group, as shown in Table 6. The obtained in vivo results indicate reasonable biocompatibility of encapsulated essential oils within chitosan–collagen polymeric material as a putative biomedical wound dressing.

### 5.7. Nanoencapsulation of Peppermint Oil and Green Tea Oil

A collaboration of Yue and co-workers from the Jiangxi University of Chinese Medicine (Nanchang, China) and the Department of Pharmacy at the 908th Hospital of People’s Liberation Army (Nanchang, China) utilized chitosan-based silica nanoparticles (CS–SiNPs) to encapsulate the peppermint oil.

Mean particle size, zeta potential, polydispersity index (PDI), and contact angles for the various ratios of chitosan within chitosan-decorated silica nanoparticles samples were estimated. The mean particle size of the silica nanoparticles (SiNPs) was 116.86 nm. However, the mean particle sizes of chitosan–SiNPs (CS–SiNPs) proportionally increased (118.12–152.5 nm) with increases in chitosan concentration, which could be due to electrostatic adsorption of chitosan over the silica nanoparticle surface. Furthermore, a significant rise in zeta potential () with the addition of chitosan from −41.8 mV (SiNPs) to 42.5 mV (SiNPs with 5% of chitosan) was also observed. The alteration of zeta potential was anticipated, as numerous hydroxyl groups on silica nanoparticles possess negative charges after deprotonation [118]. In comparison, the free amino groups of chitosan are generally protonated and therefore possess positive charges [119,120]. These opposite charges on the silica nanoparticles and chitosan facilitate their homogenized adsorption.

The 1% chitosan with silica nanoparticles had a regular sphere-shaped morphology. Scanning transmission electron microscopy (STEM) coupled with energy dispersive X-ray spectroscopy (EDS) helped in mapping the elemental composition (C, N, O, and Si), suggesting a uniform spherical distribution of carbon and nitrogen from chitosan. This observation suggested an absorption of chitosan onto the negatively charged surface of silica nanoparticles and within the agreement of zeta potential results.

The authors measured the samples for their wettability, as it was required to prepare a stable Pickering emulsion. As a general statement about the wettability, “An appropriate wettability of solid particles facilitates their absorption at the oil/water interface and provides enough steric hindrance that abolishes the droplet coalescence of Pickering emulsions” [121]. The contact angle (39.4°) of SiNPs indicated its excessive hydrophilicity and an obstacle in using it for a stable Pickering emulsion [122]. However, chitosan (from 1 to 5% *w*/*w* relative silica weight%) loaded to SiNPs showed an increase in contact angle from 41.1° to 67.4°, implying the significance of chitosan loading with SiNPs as an essential component for achieving a stable Pickering emulsion. A contact angle of 90° is considered optimum to enable absorption on the oil/water interface and to have steric hindrance against the aggregation of oil droplets; therefore, the highest contact angle of chitosan-loaded SiNPs (5% *w*/*w* relative to silica weight%) was selected for preparing a Pickering emulsion and investigated for further chemical characterization [123].

The chemical characterization was performed using XRD and FTIR analyses. The SiNPs demonstrated characteristic broad peaks at 2θ of 22°, which was in agreement with the previous literature [124]. Similarly, chitosan demonstrated broad characteristic peaks at 2θ of 10.6 and 23.2°. However, a mixture of chitosan and SiNPs, when compared with chitosan-loaded SiNPs, showed similar characteristic peaks, except for a missing peak at 2θ of 11.6° for chitosan-loaded SiNPs [123]. In FTIR, chitosan powder exhibited typical peaks, such as stretching vibrations for C–H (2939.83 cm^−1^), –OH bond (3431.15 cm^−1^), and C=O of amide-I (1633.94 cm^−1^); a bending vibration of N–H of primary amine (1528.97 cm^−1^) [125]; CH_2_ bending (1383.11) and CH_3_ symmetrical deformations (1321.56 cm^−1^); and asymmetric stretching of C–O–C (1151.90 cm^−1^) [126]. For silica, the characteristic broad peaks related to O–H stretching of silanol groups (3446.53 and 1634.79 cm^−1^) [127], Si–O–Si stretching (807.02 cm^−1^), Si–O–Si bending (469.87 cm^−1^) [128,129], and vibrations of siloxane of (SiO)_n_ groups (1112 cm^−1^) [130] were observed. However, the presence of peaks for a mixture of chitosan and unloaded SiNPs for silica (3449.10 cm^−1^, 1635.60 cm^−1^, 1102.30 cm^−1^, 805.69 cm^−1^, and 470.06 cm^−1^) and chitosan (2650.63 cm^−1^ and 1489.26 cm^−1^), and a merging of certain peaks (383.11 cm^−1^ and 1321.56 cm^−1^) confirmed their presence as components in the composition. However, the appearance or disappearance of no peaks with chitosan-loaded SiNPs were observed when compared to the mixture of chitosan and unloaded SiNPs, indicated a possible explanation that the interaction within chitosan-loaded SiNPs are of an electrostatic nature [131].

The 5% chitosan-loaded SiNPs (CS–SiNPs) were used in different proportions to prepare peppermint oil Pickering emulsions (PO-PE). The estimated particle sizes (D_50_) for 0.5% and 1% of CS–SiNPs with PO-PE were found to be large enough (6.61 ± 0.31 μm and 5.42 ± 0.25 μm), making them not ideal for stabilizing Pickering emulsions, as they were susceptible to creaming after 24 h. With further increases in the CS–SiNPs proportions (1.5 and 2% relative to PO-PE), a decrease in particle size (D_50_) was observed. The 2% CS–SiNPs/PO-PE exhibited a particle size (3.73 ± 0.213 μm) that did not show creaming during storage, indicating an efficient absorption of CS–SiNPs onto the oil–water interface of Pickering emulsions droplets. Furthermore, a sphere-shaped oil droplet morphology was observed by confocal laser scanning microscopy and cryo-SEM, indicating peppermint oil encapsulation into the core of Pickering emulsions. Additionally, these images showed silica nanoparticles at the surface of droplets, leading to speculation that CS–SiNPs make a steric barrier shell and therefore improve the stability of PO-PE [123].

Different concentrations of hydroxypropyl methyl cellulose (HPMC) was used to encapsulate the prepared material (CS–SiNPs-encapsulated PO-PE) for sustained release studies [123]. The loading capacities of CS–SiNPs-encapsulated PO-PE with HPMC (50 wt%, 75 wt%, and 100.0 wt%) were estimated at 16.7 ± 0.9%, 28.4 ± 0.7%, and 25.5 ± 1.1%, respectively. Later, the release of peppermint oil was studied with different concentrations of HPMC at varied temperatures (4, 25, 35, and 50 °C). In the initial 4 h, a slow release was noted for 25, 35, and 50 °C, with a relatively slower release for 4 °C, indicating that an increase in temperature increases the release kinetics of peppermint oil. This could arguable be because of the volatile nature of peppermint oil, which is attributed to its lower boiling point. Additionally, it was noticed that the release rate of peppermint oil from CS–SiNPs-encapsulated PO-PE was higher in lower HPMC concentrations (50 and 75%) than in 100% HPMC, directing the performance of HPMC as an encapsulating agent. This led the authors to choose the 100% HPMC-based CS–SiNPs-encapsulated PO-PE to evaluate the antibacterial activity. The disk diffusion method was used to determine the antibacterial activity, as shown in Table 7 [123].

In another study, Mamdouh’s research group from the School of Sciences and Engineering, The American University in Cairo (New Cairo, Egypt), performed a comparative study of encapsulated peppermint and green tea essential oils in chitosan nanoparticles [132]. The zeta-potentials of CSNPs with varied concentrations of peppermint essential oil (CSNPs–PE) and green tea oil (CSNPs–GTO) were found in the range of 20.9 ± 0.66 to 23.1 ± 0.4 mV and 24.2 ± 0.3 to 29.0 ± 0.2 mV, respectively. The highest zeta-potentials in both encapsulated formations (CSNPs–PO and CSNPs–GTO) were found with a 1:1 ratio; therefore, these combinations were chosen for further investigations. Additionally, the zeta potential of CSNPs was found to 24.9 ± 0.95 mV, in agreement with the previous literature [133]. Based on DLVO theory, an equilibrium between attractive van der Waals’ forces and the electrical repulsion is required. For example, a high charge density over the surface of nanoparticles abolishes their aggregation because of repulsion among them [134]. Additionally, the zeta potential value of 30.0 mV is considered ideal for stability, 20.0 mV indicates short-time stability, and ~5 mV indicates quick aggregation [135]. Although the authors did not achieve a zeta potential value of 30.0 mV for their CSNP-encapsulated essential oil samples, no aggregation was observed. This observation indicates that the zeta potential is not the only parameter that decides the stability of nanoparticles. However, a higher zeta potential of CSNPs–GTO than CSNPs–PO evidently shows a higher stability of CSNPs–GTO. A spherical particle morphology was found using TEM microscopy for 1:1 *w*/*w* ratios of CSNPs–GTO and CSNPs–PO with a size range of 20–60 nm. Furthermore, estimations of average particle sizes of CSNPs, CSNPs–PO, and CSNPs–GTO were found to be 36.1 ± 0.88 nm, 43.5 ± 1.97 nm, and 30.7 ± 1.13 nm, respectively [132]. Later, the authors determined the encapsulation efficiency (EE%) and loading capacity (LC%) for both essential oils in their nanoencapsulated forms with the help of the following equation:$$\mathrm{EE}\;\left(\%\right)=\frac{\mathrm{Total}\;\mathrm{amount}\;\mathrm{of}\;\mathrm{Loaded}\;\mathrm{Essential}\;\mathrm{oils}}{\mathrm{Initial}\;\mathrm{Amount}\;\mathrm{of}\;\mathrm{Loaded}\;\mathrm{Essential}\;\mathrm{oils}}\times 100$$
$$\mathrm{LC}\;\left(\%\right)=\frac{\mathrm{Total}\;\mathrm{amount}\;\mathrm{of}\;\mathrm{Loaded}\;\mathrm{Essential}\;\mathrm{oils}}{\mathrm{Weight}\;\mathrm{of}\;\mathrm{nanoparticles}\;\mathrm{after}\;\mathrm{Freeze}\;\mathrm{drying}}\times 100$$

The loading capacities (LC%) of the encapsulated peppermint oil and green tea oil were found in the range of 8.15–22.2% and 2.2–23.1%, respectively. Additionally, the encapsulation efficiencies (EE%) of CSNPs–PO and CSNPs–GTO were determined as 78–82% and 22–81%, respectively, in agreement with the previous literature [106].

A 72 h in vitro release study of CSNPs–PO and CSNPs–GTO was performed in buffer (pH = 3 (acetate buffer) and 7.4 (phosphate-buffered saline)). After an initial 12 h period phase, 45.7% and 74.5% for peppermint essential oil and green tea oil, respectively, were released in acidic acetate buffer, while a slow release was noticed in acidic acetate buffer until 72 h (61.3% and 74.9% for peppermint essential oil and green tea oil, respectively). Similar observations were also made with saline buffer (pH = 7.4), where the initial 12 h release (35.8% and 57.4% for peppermint essential oil and green tea oil, respectively) and until 72 h (50.7% and 62.9% for peppermint essential oil and green tea oil, respectively) were recorded. In our opinion, based on the results reported by the authors, a more controlled in vitro release of green tea oil than peppermint oil was observed in both buffers after 12 h [132].

The agar dilution and colony counting methods were used to evaluate the antibacterial activity of peppermint oil and green tea oil before and after their encapsulation with CSNPs. The antibacterial activity was measured as minimum bactericidal concentration (MBC), which is defined as “the lowest concentration of sample/agent that kills 99.9% or more of the initial inoculum”. The bacteria used in antibacterial assays were *Staphylococcus*
*aureus* (as a representative Gram-positive bacteria) and *Escherichia coli* (as a representative Gram-negative bacteria). In both cases, encapsulated essential oil samples were found with more anti-staphylococcal activity (MBC values: CSNPs–PO = 1.11 mg·mL^−1^, versus peppermint oil = 1.36 mg·mL^−1^; CSNPs–GTO = 0.57 mg·mL^−1^, versus green tea oil ≥ 5.44 mg·mL^−1^). On the contrary, CSNPs exhibited 5.0 mg·mL^−1^ of anti-staphylococcal activity. These results exemplify an example of the decisive role of chitosan nanoencapsulation in enhancing the antibacterial activity of essential oil [132].

Antibacterial testing on *Escherichia coli* (MBC values: CSNPs–PO ≥ 2.72 mg·mL^−1^, versus peppermint oil = 2.72 mg·mL^−1^; CSNPs–GTO = 1.15 mg·mL^−1^, versus green tea oil = 5.44 mg·mL^−1^) exhibited a similar trend in the case of green tea oil. The antibacterial (*Escherichia coli*) activity for CSNPs was measured at 7.50 mg·mL^−1^ [132].

### 5.8. Nanoencapsulation of Cardamom Oil

Jamil et al. encapsulated cardamom essential oil in chitosan nanocomposites. In their study, six essential oils were evaluated for antibacterial activity (cardamom, lemon, rose, peppermint, eucalyptus, and orange). Methicillin-resistant *Staphylococcus*
*aureus* (MRSA) and *Escherichia coli* bacteria were studied for antibacterial activity using the agar well diffusion method. A prescreening of antibacterial activity was performed at 50 µL, where cardamom was found to have the most antibacterial activity against MRSA and *Escherichia coli*. Later, CSNPs were prepared using the conventional ionic gelation process. By SEM and AFM, a size range of 50–100 nm for CSNPs was measured. The cardamom oil-loaded CSNPs were evaluated using FTIR data. A zeta potential value of 50.0 mV was found in unloaded CSNPs as well as in cardamom oil-loaded CSNPs.

Later growth kinetic studies were performed for *Escherichia coli* and MRSA. Both samples (CSNPs and cardamom oil-loaded CSNPs) exhibited control over bacteria growth until 48 h. Surprising, after 48 h, the CSNPs were found to be ineffective, while more control was observed with cardamom oil-loaded CSNPs, which lasted for 7 days. However, these results did not shed enough light to explain why such events took place after 48 h, and the authors must investigate this further [136].

### 5.9. Nanoencapsulation of Clove Essential Oil

Hadidi et al. reported clove essential oil-loaded CSNPs as antibacterial material for *Listeria monocytogenes* and *Staphylococcus aureus*. The authors extracted the clove essential oil, where GC–MS was used to characterize the 23 compounds. The major components of essential oil were eugenol (89.86%) and β-caryophyllene (5.40%), which is in agreement with the previous literature [137,138]. Various concentrations of essential oil loading were used with chitosan. The clove essential oil-loaded CSNP ratios were evaluated for their physicochemical characterization, as shown in Table 8.
$$\mathrm{Retention}\;\mathrm{of}\;\mathrm{Essential}\;\mathrm{oil}\;\left(\%\right)=\frac{\mathrm{Total}\;\mathrm{amount}\;\mathrm{of}\;\mathrm{Loaded}\;\mathrm{Essential}\;\mathrm{oils}}{\mathrm{Initial}\;\mathrm{Amount}\;\mathrm{of}\;\mathrm{Loaded}\;\mathrm{Essential}\;\mathrm{oils}}\times 100$$

The chemical characterization was performed with FTIR. The unloaded CSNP showed characteristic peaks at 3445 cm^−1^ (O–H), 3298 cm^−1^ (N–H_2_ stretching), 2991 cm^−1^ (C–H stretching), 1546 cm^−1^(–CONH_2_ of amide-II), 1367 cm^−1^ (C–N stretching), 1201 cm^−1^ (β-(1−4) glycosidic linkage), 1065 cm^−1^ (C–O–C stretching of glucose ring), 997 cm^−1^ (C–O stretching), and 904 cm^−1^ (vibration of the pyranose ring). Surprisingly, C=O stretching of amide-I and the other two peaks (1065 cm^−1^ of C–O–C stretching of glucose ring; 1545 cm^−1^ of amide-II) did not appear, suggesting ionic crosslinking between the PO_4_^3−^ group of TPP and NH^3+^ group of chitosan [94,107]. However, the increase in the intensity of the C–H stretching (2991 cm^−1^) peak for clove essential oil-loaded CSNPs suggested encapsulation [139].

The inhibition halo (cm) of clove essential oil CSNPs (CSNP:essential oil ratio of 1:0.5) on *Staphylococcus*
*aureus*, *Listeria monocytogenes*, *Salmonella Typhi*, and *Escherichia coli* for 32 μL of minimum inhibitory volume (MIV), was found to be 4.8, 4.78, 4.49, and 3.95 cm, respectively [139].

### 5.10. Nanoencapsulation of Cumin Seed Essential Oil

Karimirad et al. from the Faculty of Agriculture and Natural Resources, University of Mohaghegh Ardabili (Ardabil, Iran), extracted essential oil from cumin seeds [140]. The GC–MS analysis showed six monoterpenes as major compounds of essential oil (cuminic aldehyde 23.6%, γ-terpinen-7-al 22.23%, γ-terpinene 19.2%, β-pinene 15.4%, p-cymene 7.16%, α-terpinen-7-al 6.84%). The diameter and distribution of CSNPs and cumin seed essential oil (CSEO)-loaded CSNPs were determined by DLS. The results indicated that the mean sizes of CSNPs and CEO–CSNPs ranged from 39.06 to 51.75 ± 0.121 nm and 41.69 to 52.77 ± 5.200 nm, respectively. An analysis of morphology and size of CEO–CSNPs by transmission electron microscopy (TEM) revealed that CEO–CSNPs had spherical shapes with diameters ranging from 30 to 80 nm. Further analysis of CSNP:cumin seed essential oil (1:0.25) showed encapsulation efficiency (=17.89 ± 0.054%), loading capacity (6.88 ± 0.021%), particle size (=52.77 ± 5.20 mm), and polydispersity index (=0.181). When tested against the bacteria (mesophiles, psychrophiles) after 20 days of storage, they were 9.01 ± 0.098 and 9.26 ± 0.10 log10 cfu/g, respectively [140].

### 5.11. Nanoencapsulation of Carvacrol

Keawchaoon et al. from the Department of Packaging and Materials Technology, Faculty of Agro-Industry, Kasetsart University (Bangkok, Thailand), studied the chitosan-based nanoencapsulation of carvacrol. Carvacrol is considered as a safe food additive and is mainly derived from the essential oils of marjoram, oregano, summer savory, and thyme [141,142]. Although carvacrol has broad applications in the cosmetic, drug, and food industries, it is sensitive towards heat, light, and oxygen. To enhance its stability and shelf-life, the authors attempted a nanoencapsulation strategy. A two-step process (droplet formation and droplet solidification) was used to prepare the carvacrol-loaded CSNPs. The oil-in-water emulsion technique was implemented for carvacrol droplet formation in chitosan solution, while droplet solidification was conducted by the cross-linking of polyphosphate groups (P3O105−) of TPP molecules with protonated NH_2_ groups (NH_3_^+^) of chitosan molecules enclosing the carvacrol droplet [106].

Chemical characterization was performed by FTIR. Carvacrol showed peaks at 3378 (–OH); 2960 (C–H stretching); 1459, 1382, and 1346 (C–H deformation); and 866 and 812 cm^−1^ (aromatic ring) [106]. CSNP peaks were found at 3500–3250 (–OH), 2927 (C–H stretching), 1634 (–CONH_2_, amide-I), 1539 (–CONH_2_, amide-II), 1155 (P=O) [70,143,144], 1072 (C–O–C), and 890 cm^−1^ (pyranose ring). There were no substantial differences found in carvacrol-loaded CSNPs, except the pronounced increase in the C–H stretching peak at 2870–2959 cm^−1^. This evidently provided a clue of carvacrol presence in the chitosan matrix.

In TGA analysis, mass losses of samples were studied as functions of temperature and to evaluate the thermal stability of the respective samples. During this study, a DTG thermogram was plotted, where the decomposition temperature (T_d_) represented a peak as a corresponding temperature to a maximum mass loss of sample [106]. Carvacrol showed a one-step mass loss from 183.8 °C (peaks at 213.9 °C), while CSNPs showed a two-step mass loss from 90.5 °C and 231.3 °C. A two-step mass loss in CSNPs reflected moisture evaporation followed by the decomposition of chitosan. However, carvacrol-loaded CSNPs showed two new T_d_ values (183.5–186.4 °C; 322.2–340.6 °C), indicating the consequent loss of free carvacrol and encapsulated carvacrol. Interestingly, the significant increase of T_d_ values of carvacrol to carvacrol-loaded CSNPs showed an enhancement of thermal stability.

The encapsulation efficiency (EE) and loading capacity (LC) of carvacrol-loaded CSNPs were evaluated for various combinations of carvacrol with CSNPs (CSNP:carvacrol = 1:0, 1:0.25, 1:05, 1.0.75, 1:1, 1:1.25). A proportional increasing encapsulation efficiency trend was observed up to a 1:1 ratio (EE = 31.4 ± 1.3), which could be CSNPs saturated with carvacrol. For the ratio of 1:1 of CSNP:carvacrol loading capacity (from UV–Vis spectroscopy = 18.9 ± 0.8; from TGA analysis = 18.9; from FTIR, as a comparison of intensity peak ratio at 2959 (for C–H stretching) and 890 (for pyranose ring) = 1.6), Z-average diameter (= 695.9 ± 48.8 nm) and zeta potential (= 29.3 ± 0.9 mV) were estimated.
$$\mathrm{EE}\;\left(\%\right)=\frac{\mathrm{Mass}\;\mathrm{of}\;\mathrm{loaded}\;\mathrm{Carvacrol}}{\mathrm{Mass}\;\mathrm{of}\;\mathrm{Initial}\;\mathrm{Carvacrol}}\times 100$$
$$\mathrm{LC}\;\left(\%\right)=\frac{\mathrm{Mass}\;\mathrm{of}\;\mathrm{loaded}\;\mathrm{Carvacrol}}{\mathrm{Mass}\;\mathrm{of}\;\mathrm{Sample}}\times 100$$

Later, carvacrol-loaded CSNPs (1:1) were tested against the three strains of bacteria (*Escherichia coli*, *Staphylococcus aureus*, and *Bacillus cereus*) using a broth dilution assay. Unloaded CSNPs when evaluated for antibacterial activity were found to be 8.225 mg/mL and unable to prevent the growth of these three bacteria strains. However, there were minimum bactericidal concentrations (MBCs) for carvacrol-loaded CSNPs of *Staphylococcus aureus* (= 4.113 mg/mL), *Bacillus cereus* (= 2.056 mg/mL), and *Escherichia coli* (= 8.225 mg/mL).

In another study, Mexican researchers (collaboration of Universidad de Sonora and Centro de Investigación en Alimentación y Desarrollo) nanoencapsulated the carvacrol into CSNPs against *Pseudomonas aeruginosa* biofilms [145]. The carvacrol-loaded CSNPs prevented the growth of *Pseudomonas aeruginosa* in biofilms (0.078–2.0 log CFU·cm^−2^) and reduced the swarming motility (40–60%). Additionally, reduced quorum sensing in *Chromobacterium violaceum* was observed [145].

### 5.12. Nanoencapsulation of Lemongrass Essential Oil

A research collaboration (from College of Agriculture, University of Tabriz (Tabriz, Iran); Tabriz University of Medical Sciences (Tabriz, Iran); Parque Tecnológico de Galicia, (Ourense, Spain); Universidad de Vigo (Ourense, Spain)) encapsulated lemongrass (*Cymbopogon commutatus*) essential oil into chitosan nanoparticles (CSNPs) [146]. Initially, the authors extracted the lemongrass oil and ran GC–MS analysis. GC–MS analysis showed 38.64% of α-citral (geranial), 30.34% of β-citral (neral), 8.22% of geranyl acetate, and 6.31% of β-myrcene, which were present in high concentrations.

Various combinations of CSNPs and essential oil were used (1:0, 1:0.25, 1:0.50, 1:0.75, 1:1, 1:1.25) and showed a spherical morphology (an average hydrodynamic size of 175–235 nm for 1:0.75 *w*/*w*). The encapsulation was accessed with the help of UV–Vis (encapsulation efficiency percentage and loading percentage) and FTIR spectroscopy (relative comparison of peaks appeared at 2925 cm^−1^ and 891 cm^−1^ between essential oil-loaded CSNPs and unloaded CSNPs). By UV–Vis spectroscopy, the (1:0.75) ratio of CSNPs with essential oil showed the highest encapsulation efficiency percentage (44.82 ± 2.80%) among other ratios. A similar pattern was also obtained for FTIR spectroscopy (a value of 4.86). Furthermore, the loading capacity of the ratio (1:0.75) was found to be 16.10 ± 1.10% [146]. The antibacterial testing was measured by minimum inhibitory concentration (MIC), minimum microbicidal concentration (MMC), and diameter of inhibition zone (DIZ) on *Escherichia coli*, *Listeria monocytogenes*, and *Staphylococcus*
*aureus*, as shown in Table 9 [146].

### 5.13. Nanoencapsulation of Summer Savory Essential Oil

Feyzioglu et al. from Yildiz Technical University, Faculty of Chemical and Metallurgical Engineering, Department of Food Engineering (Istanbul, Turkey), extracted the essential oil from leaves of summer savory (*Satureja hortensis* L.) [147]. Various percentage of extracted essential oil were used (1%, 1.2%, 1.4%, 1.5%) with CSNPs. However, the zeta potential and encapsulation efficiency (%) were found with uneven trends from −7.54 to −7.54 mV, and from 35.07 to 40.70%, respectively. The antibacterial activity of essential oil-loaded CSNP were tested against the *Escherichia coli* O157:H7, *Listeria monocytogenes*, and *Staphylococcus*
*aureus*. All essential oil-loaded CSNPs showed antibacterial activity against *Escherichia coli* O157:H7 and *Listeria*
*monocytogenes*, but a moderate effect was found against *Staphylococcus*
*aureus* [147].

### 5.14. Nanoencapsulation of Thyme Essential Oil

Mexican researchers (collaborative work from the Instituto Politécnico Nacional-Centro de Desarrollo de Productos Bióticos (CeProBi-IPN); CONACYT.Instituto Politécnico Nacional-Centro de Desarrollo de Productos Bióticos (CeProBi-IPN) and Instituto Politécnico Nacional-Unidad Profesional Interdisciplinaria de Biotecnología (UPIBI-IPN)) nanoencapsulated thyme essential oil (TEO) into chitosan nanoparticles (TEO–CSNPs) and nanocapsules (TEO-CSNCs) [148]. Through TEM microscopy, the average size of TEO–CSNPs was estimated to be 6.4 ± 0.5 nm, and for TEO-CSNCs it was estimated to be 9.1 ± 1.6 nm. The encapsulation efficiency for thymol (TEO–CSNPs = 68 ± 1%; TEO–CSNCs = 72 ± 1%) and for carvacrol (TEO–CSNPs = 80.5 ± 1%; TEO–CSNCs = 81.4 ± 1%) was found to be in agreement with the previous literature [149].

The antibacterial activity of TEO–CSNPs and TEO-CSNCs was evaluated against *Staphylococcus*
*aureus*, *Listeria monocytogenes*, *Bacillus cereus*, *Salmonella typhi*, *Shigella dysenteriae*, and *Escherichia coli* at different concentrations (2.5, 5, 10, 20, and 40 μL). Generally, a significantly higher inhibitory activity was observed for TEO–CSNPs than TEO-CSNCs [148].

### 5.15. Nanoencapsulation of Cinnamomum zeylanicum Essential Oil

Iranian Researchers (collaborative work from the Department of Microbiology, Shahid Beheshti University; Microbial Biotechnology and Biosafety Department, AREEO) nanoencapsulated *Cinnamomum* essential oil into chitosan nanoparticles [150]. The *Cinnamomum* essential oil loading to CSNPs was carried out in *w*/*w* ratios (CSNPs:essential oil (*w*/*w*) = 1:0, 1:0.25, 1:0.50, 1:0.75, 1:1). Interestingly, a decreasing trend of encapsulation efficiency was observed as the loading of essential oil increased (CSNPs:essential oil (*w*/*w*) =1:0.25, with the highest encapsulation efficiency of 16.91 ± 0.51%). Similar values were achieved for loading capacity for all the essential oil samples. Furthermore, essential oil-loaded CSNPs showed antifungal activity [150]. Although this study is not in direct comparison to antibacterial activity, it shows a possible use of *Cinnamomum zeylanicum* essential oil-loaded CSNPs as antibacterial materials.

### 5.16. Encapsulation of Eucalyptus Oil Nanoemulsion

Sugumar et al., from the Centre for Nanobiotechnology, VIT University (Vellore, India), prepared an *oil-in-water* nanoemulsion using eucalyptus oil (6%) (*Eucalyptus globulus*), nonionic surfactant (Tween 80; Tween 20), and water (1:2 *v*/*v*) [151]. A chitosan matrix was used to encapsulate the different percentages of eucalyptus oil nanoemulsion (0%, 1%, 3%, and 5% *v*/*v*). The anti-streptococcal activity (*Staphylococcus*
*aureus*) was performed with essential oil-loaded (1%, 3%, and 5%) chitosan samples using the agar disc diffusion method. Then inhibition zones were measured for these essential oil-loaded (0, 1%, 3%, and 5%) chitosan samples (7 ± 0.1, 7 ± 0.5, 11 ± 0.3, 15 ± 1 mm), respectively [151].

### 5.17. Nanoencapsulation of Nettle Essential Oil

Bagheri et al. used chitosan nanoencapsulation for nettle essential oil. The authors extracted the essential oil from nettle leaves (*Urtica dioica* L.). The chemical composition of nettle essential oil showed carvacrol (40.6%), carvone (10.5%), naphthalene (9.8%), (E)-anethol (3.9%), and (E)-α-ionone (3.1%), which is in agreement with the previous literature [152]. A two-step methodology was used to encapsulate the nettle essential oil into CSNPs: *oil-in-water* emulsification followed by ion gelation. Based on high retention (68.2 ± 3.1%) of nettle essential oil, a ratio of CSNPs to essential oil of 1:0.5 was selected for further investigations. Additional physiochemical characterization showed particle size (273.8 ± 39.2 nm), zeta potential (+20.33 ± 1.1mV), and polydispersity index (0.255 ± 0.023). The antibacterial evaluation was performed on four bacteria strains, namely, *Escherichia coli*, *Bacillus cereus*, *Salmonella*
*typhi*, *Listeria*
*monocytogenes*, and *Staphylococcus*
*aureus*, where inhibition halo values were 4.11, 3.53, 3.46, 3.95, and 3.45 cm, respectively [153].

## 6. Conclusions

Natural biopolymers always present a choice of platforms to enhance the material properties of other functional materials. Various natural polysaccharides (alginic acid, sulfated polysaccharides, hyaluronic acid, κ-carrageenan, pectin and dextran, and chitosan) were found with reliable mechanical and material properties. However, the distinctive chemical reactivity of the chitosan structure compared to other polysaccharides because of the presence of NH_2_ groups makes it more attractive among material chemists. For example, the NH_2_ chemical heads on C-2 of the glycosamine unit on chitosan can easily be protonated, yielding polycationic chitosan. Interestingly, bacterial membrane contains a negative charge; polycationic chitosan can interact through ionic–ionic charges and behave as an antibacterial material [33]. Secondly, reducing its particle size further amplifies its antibacterial role [154]. Therefore, nanoscale-size particles can be used for nanoencapsulation-based formulations along with antibiotics or other natural antibacterial compounds. The nanoparticle-based formulations have already found various applications: drug-delivery carrier [155], selectivity-controlled catalysis [156], smart materials [157], etc.; therefore, various chemical processes are commonly practiced and optimized.

One estimation showed that 3000 essential oils are of plant origin, and approximately 300 are commercially important, with a market value over $700 million US [158]. However, essential oils have shown persistent antibacterial activities [159], but their high hydrophobicity and tendency to vaporize at room temperatures limit their direct use in material-making applications. However, encapsulation with chitosan improves their limited physicochemical properties and enhances their antibacterial nature. One could argue that FDA-approved antibiotics (such as tetracyclines, macrolides, etc.)-based encapsulation with chitosan could serve better as antibacterial materials, but a higher cost and large molecular sizes (molecular obesity) of antibiotics restrict their practicality. The second issue with antibiotics-based encapsulation is that their specific physicochemical properties restrict their uniformity in functionalization over materials. However, the microbial ecology of material surfaces contains biofilms but not planktonic bacteria; therefore, using antibiotics to target bacteria of biofilms can severely increase the chances of antibiotic resistance, which can further jeopardize public health. The microbes live in syntropy in these biofilms, therefore continuously exchanging the genetic materials, and show robustness towards antibiotic treatment; for example, 72,000 deaths in the United States alone were reported in 2015. Therefore, nanoencapsulation of phytochemicals is becoming an increasing trend.

The choice of nanoencapsulation is ideal, as various applications have already been devised—enhancing the delivery of a specific drug or protein inhibitor to its target site; reducing the research cost burden on drug companies during the preclinical development of drugs [160,161]; minimizing the on-target/off-target toxicity profile of inhibitors/drugs [162,163]; providing the in vivo metabolic stability of medicinally active compounds [164,165]; and providing catalytic support (such as Cu (II) on magnetic chitosan for substituted pyridines [166,167] or catalytic stability [168] for light-/chemo-sensitive molecular heads, such as azobenzene [169,170] and spiro compounds [171]. However, choosing an alternative polymeric matrix would also increase the shelf-life of essential oils or their major aromatic constituents. One example is phosphatidylcholine-based liposomes, which could be used to nanoencapsulate essential oils (carvacrol) that prevent oxidative degradation [111], but the instability of liposomes is again a concern. Therefore, the researchers opted for the chitosan supporting matrix to encapsulate the essential oils.

## Figures and Tables

**Figure 1 micromachines-13-01265-f001:**
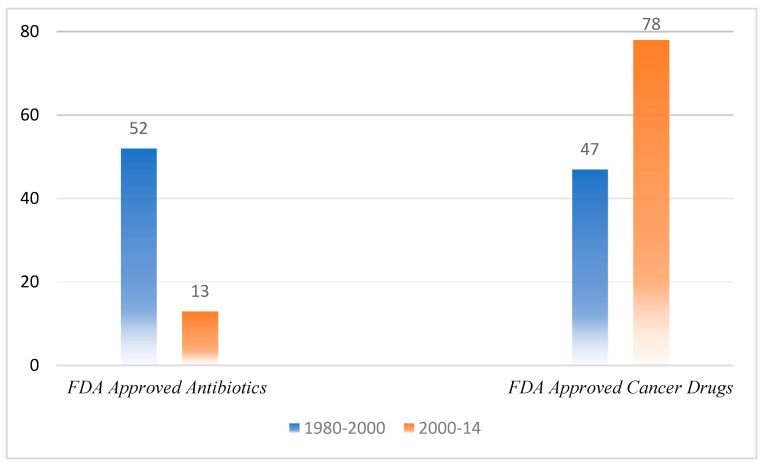
Illustration of FDA-approved drugs (antibiotics vs. anticancer) in the last 40 years.

**Figure 2 micromachines-13-01265-f002:**
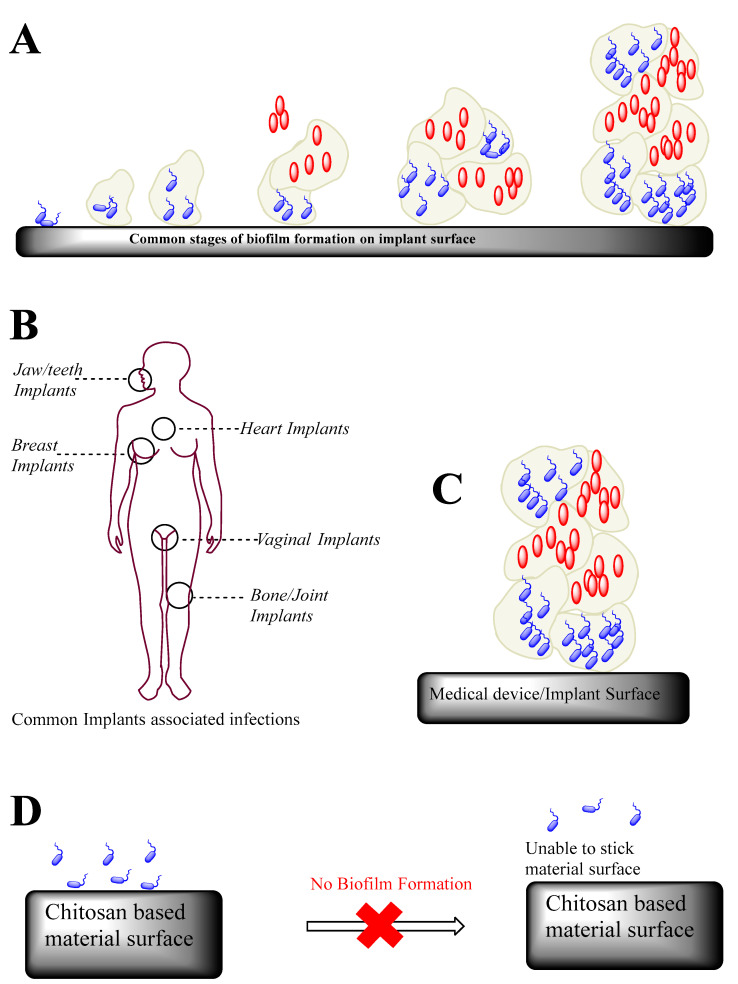
(**A**) Formation of microbial biofilm on the surface of implants. (**B**) Various human implants for biomedical use. (**C**) Biofilm formation on implant surface showing different microbial colonies. (**D**) Application of chitosan-based material surfaces to prevent microbial biofilm formation [11].

**Figure 3 micromachines-13-01265-f003:**
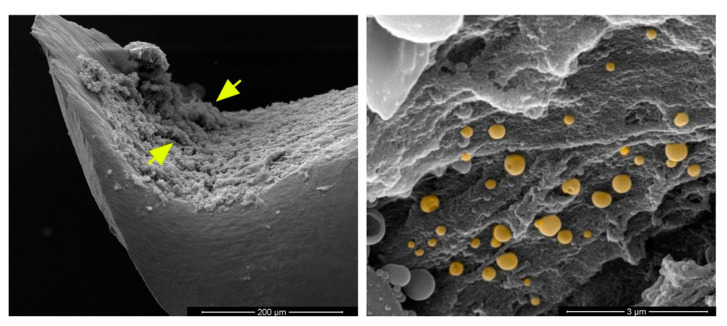
Scanning electron microscopy: Biofilm on the surface of an orthopedic implant [12]. adopted from Heim et al. [12].

**Figure 4 micromachines-13-01265-f004:**
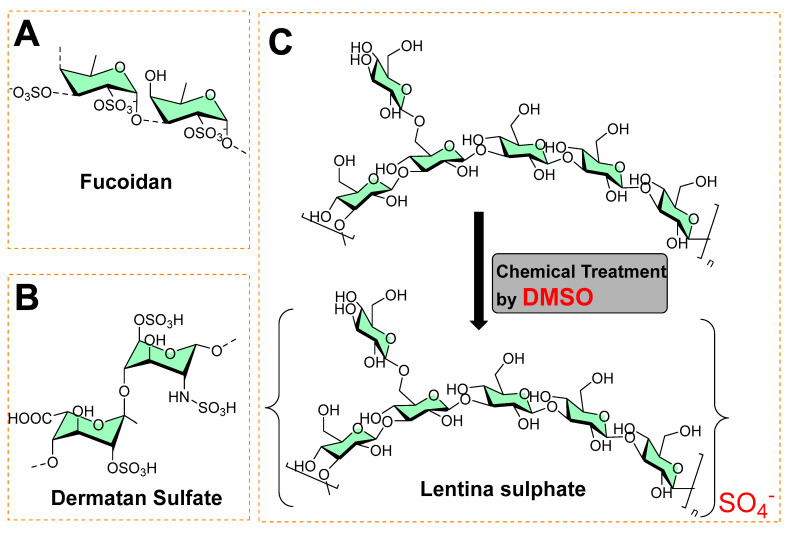
(**A**) Chemical structure of fucoidan. (**B**) Chemical structure of dermatan sulfate. (**C**) Chemical synthesis of lentina sulfate using DMSO.

**Figure 5 micromachines-13-01265-f005:**
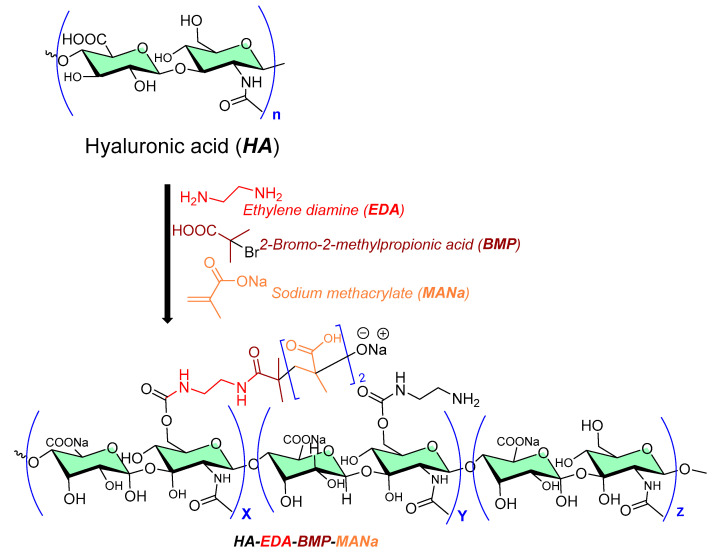
Chemical processing of hyaluronic acid copolymer as antibiofilm surface material.

**Figure 6 micromachines-13-01265-f006:**
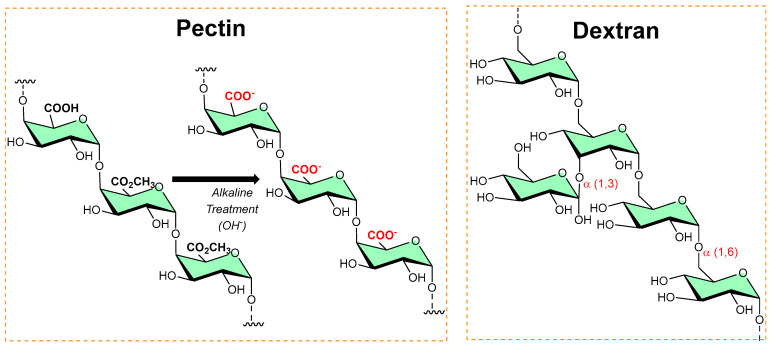
The typical chemical process of converting pectin into a more functional form, while the dextran structure shows alpha linkages and available functionalized sites.

**Figure 7 micromachines-13-01265-f007:**
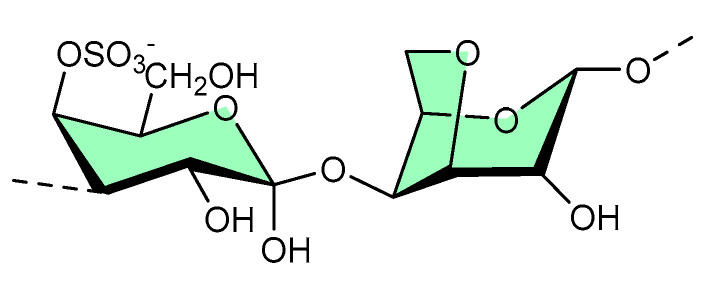
Available site for material functionalization on κ-carrageenan structure.

**Figure 8 micromachines-13-01265-f008:**
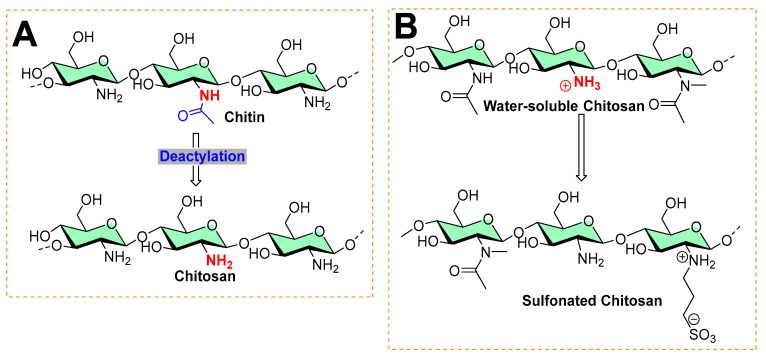
Typical processes in chitosan chemistry: (**A**) commercial preparation from chitin by deacetylation is found in the exoskeleton of crustaceans and the cell wall of fungi. (**B**) Sulfonation of chitosan.

**Figure 9 micromachines-13-01265-f009:**
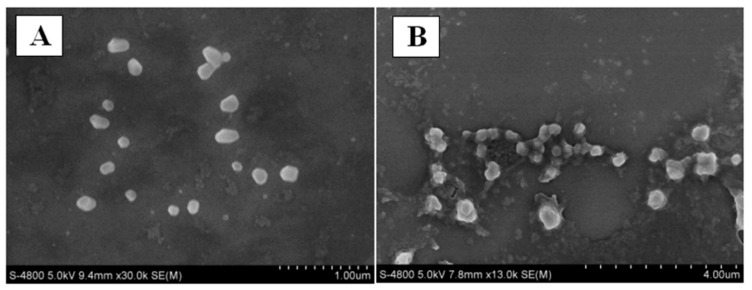
Scanning electron microscopy images of CSNPs (**A**) without loading of BEO, and (**B**) with a loaded ratio of 1:0.5 with BEO. Reproduced with permission from Cai et al. [93]. Copyright 2022 Elsevier.

**Figure 10 micromachines-13-01265-f010:**
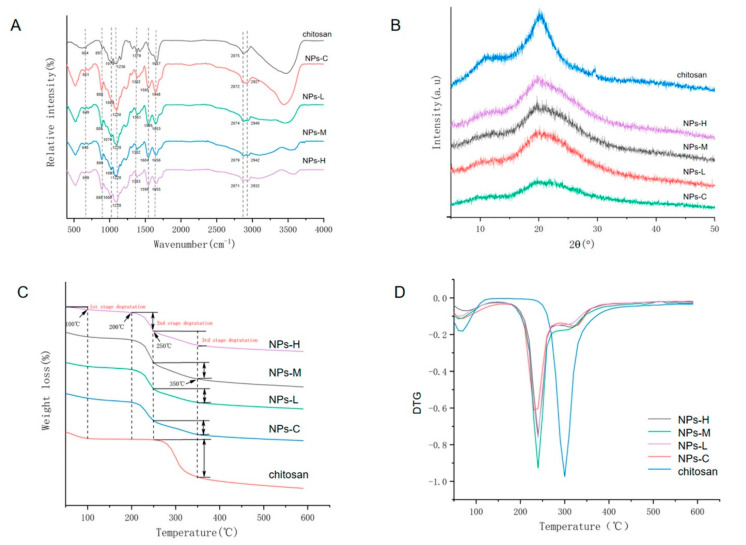
(**A**) FTIR results showing chitosan. (**B**) XRD analysis (**C**), TGA analysis (**D**), DTG analysis. The ratios of chitosan to MEO: (NPs-C = 1:0), NPs-L (1:0.25), NPs-M (1:0.5), NPs-H (1:1). Reproduced with permission from Song et al. [96]. Copyright 2021 Elsevier.

**Figure 11 micromachines-13-01265-f011:**
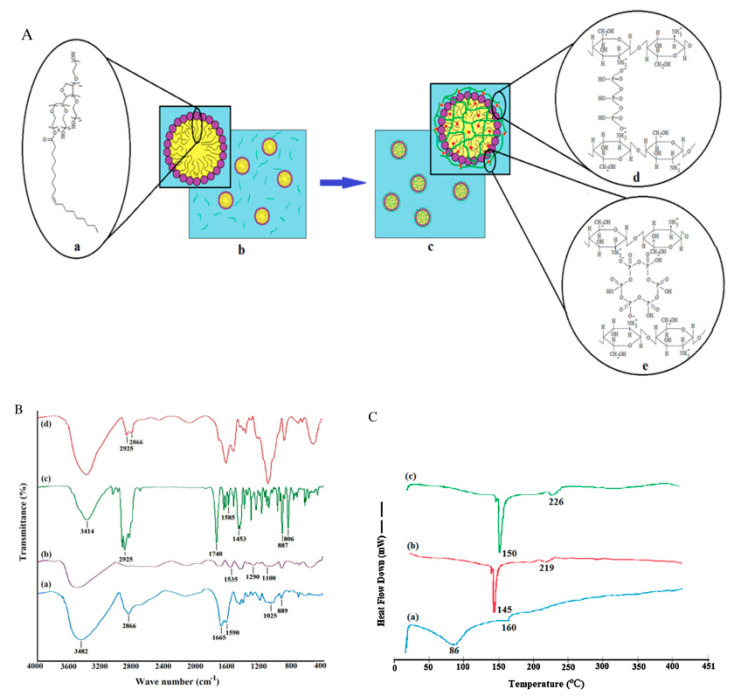
(**A**) Schematic representation of the chemical structure of Tween 80 (a), CEO in chitosan droplets (*oil-in-water* emulsion) (b), CEO-loaded chitosan particles (c), the chemical structure of chitosan ionically cross-linked with TPP (d), and chemical structure of chitosan ionically cross-linked with HMP (e). (**B**) FTIR spectra of chitosan (a), chitosan–TPP nanoparticles (b), CEO (c), and CEO-loaded nanoparticles with a chitosan to CEO weight ratio of 1:0.25 (d). (**C**) DSC curves of CEO (a), chitosan–TPP nanoparticles (b), and CEO-loaded chitosan nanoparticles (c). Reproduced with permission from Esmaeili et al. [101]. Copyright 2015 Elsevier.

**Figure 12 micromachines-13-01265-f012:**
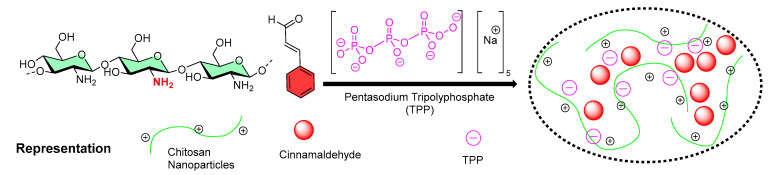
Schematic representation of the preparation of encapsulation of cinnamaldehyde (CA) into CSNPs.

**Figure 13 micromachines-13-01265-f013:**
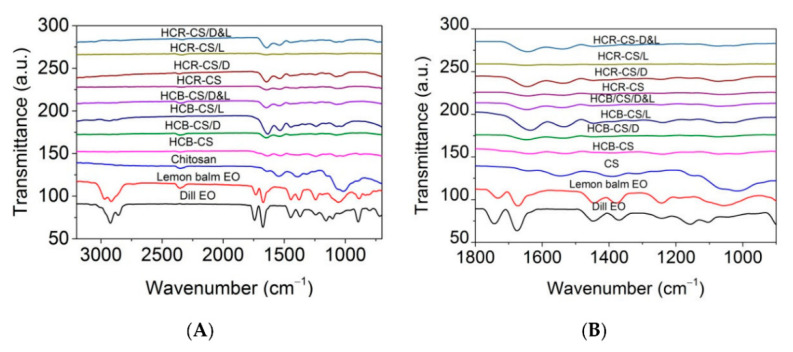
Comparative FTIR data of various samples between 4000 and 700 cm^−1^ (**A**); zoomed in range of 1800 and 900 cm^−1^ (**B**). Abbreviations and represented names in the figure as follows: dill EO (AGEO), lemon balm EO (MOEO), HCB-CS (mixture of collagen hydrolysates of bovine source (HCB), and chitosan (CS)), HCB-CS/D (mixture of collagen hydrolysates of bovine source (HCB) and chitosan (CS) along with the AGEO), HCB-CS/L (mixture of collagen hydrolysates of bovine source (HCB) and chitosan (CS) along with the MOEO), HCB-CS/D&L (mixture of collagen hydrolysates of bovine source (HCB) and chitosan (CS) along with the AGEO and MOEO), HCR-CS, HCR-CS/D, HCR-CS/L, HCR-CS/D&L. Reproduced from Rapa et al. [116] under a Creative Common CC BY license.

**Figure 14 micromachines-13-01265-f014:**
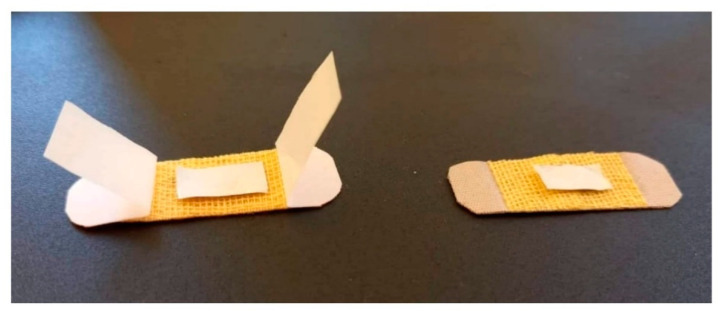
Photograph of prepared nanofiber samples before animal testing. Reproduced from Rapa et al. [116] under a Creative Common CC BY license.

**Table 1 micromachines-13-01265-t001:** Chitosan nanoparticle preparation methods [59].

Preparation Method	Matrix Composition	Reference
Polyelectrolyte complexation	CS, alginate, Arabic gum, carboxymethyl cellulose, carrageenan, chondroitin sulfate, cyclodextrins, dextran sulfate, polyacrylic acid, poly-γ-glutamic acid, insulin, DNA	[60,61,62,63,64,65,66,67,68,69,70,71,72]
Ionic gelation	CS, tripolyphosphate	[73,74,75,76,77]
Emulsification and cross-linking	CS, glutaraldehyde	[78,79]
Emulsion droplet coalescence	CS	[80,81]
Desolvation	CS	[82,83,84,85,86]
Reverse micellization	CS, glutaraldehyde	[87,88,89,90]
Modified ionic gelation with radical polymerization	CS, acrylic acid, methacrylic acid, polyethylene glycol, polyether	[61,91,92]
Emulsion solvent diffusion	CS	[82,83,84,85,86]

CS = chitosan.

**Table 2 micromachines-13-01265-t002:** Characterization of FTIR spectra of chitosan powder with chitosan-TPP [102].

Chitosan Powder	Chitosan–TPP	Inferences
3482 (–OH and NH_2_ stretching)		
2866 (–CH stretching)		
1665 (amide-I)		
1590 (amide-II, NH bending)	1535 (amide-II)	*Shift in peak*
1025 (C–O–C stretching)		
889 cm^−1^ (glucose ring)		
	1100–1290 (P–O and P=O)	*New peak originated*

**Table 3 micromachines-13-01265-t003:** Comparison of antibacterial activity of non-encapsulated CEO with CSNPs.

Materials	Antibacterial Activity (Inhibition Zone, mm)					
	Gram-Positive Bacteria			Gram-Negative Bacteria		
	*S. aureus*	*S. epidermidis*	*B. cereus*	*E. coli*	*S. typhimurium*	*P. vulgaris*
CEO	9.7 ± 1.0	9.3 ± 0.8	8.8 ± 0.3	7.8 ± 0.3	6.8 ± 0.3	7.2 ± 0.8
DMSO (negative control)	0.0	0.0	0.0	0.0	0.0	0.0
CEO-loaded CSNPs	11.3 ± 1.0	12.3 ± 1.0	10.7 ± 0.3	10.0 ± 1.0	9.5 ± 1.3	8.0 ± 1.0
PBS (negative control)	0.0	0.0	0.0	0.0	0.0	0.0
CSNPs	6.7 ± 0.8	9.0 ± 0.5	7.0 ± 0.5	6.7 ± 0.3	6.2 ± 0.8	5.8 ± 0.8
PBS (negative control)	0.0	0.0	0.0	0.0	0.0	0.0

**Table 4 micromachines-13-01265-t004:** Enlisting the results obtained from diffusion disk assay: represented by the diameters of the inhibition zones (measured in mm).

Sample	*S. aureus*	*E. coli*	*E. faecalis*	*S. typhimurium*	*C. albicans*	*C. glabrata*	*A. brasiliensis*
AGEO	8.94 ± 0.04	-	-	-	-	16.34 ± 0.14	-
HCB-CS	12.94 ± 0.31	17.21 ± 0.04	-	17.47 ± 0.11	18.29 ± 0.28	22.50 ± 0.34	23.64 ± 0.27
HCB-CS/AGEO	11.19 ± 0.18	19.09 ± 0.31	16.12 ± 0.08	15.33 ± 0.35	15.69 ± 0.07	26.53 ± 0.24	16.72 ± 0.47
HCR-CS	20.67 ± 0.21	-	28.56 ± 0.23	29.88 ± 0.27	19.05 ± 0.17	16.03 ± 0.47	20.03 ± 0.08
HCR-CS/AGEO	21.40 ± 0.17	10.27 ± 0.12	26.79 ± 0.12	30.88 ± 0.13	19.60 ± 0.12	42.58 ± 0.57	16.14 ± 0.21

**Table 5 micromachines-13-01265-t005:** Results of antimicrobial testing.

Sample	*S. aureus*	*E. coli*	*E. faecalis*	*S. typhimurium*	*C. albicans*	*C. glabrata*	*A. brasiliensis*
L	9.04 ± 0.25	9.09 ± 0.12	8.42 ± 0.14	-	-	12.24 ± 0.35	-
D&L	7.86 ± 0.45	8.12 ± 0.07	7.00 ± 0.31	-	-	8.06 ± 0.18	-
HCB-CS/L	17.39 ± 0.21	25.09 ± 0.11	26.70 ± 0.12	18.87 ± 0.54	17.41 ± 0.31	22.50 ± 0.54	15.62 ± 0.32
HCB-CS/D&L	26.43 ± 0.05	22.79 ± 0.41	25.28 ± 0.51	13.19 ± 0.11	19.61 ± 0.23	30.35 ± 0.33	14.68 ± 0.22
HCR-CS/L	34.93 ± 0.07	9.46 ± 0.13	28.71 ± 0.24	27.54 ± 0.24	19.47 ± 0.05	51.12 ± 0.24	10.74 ± 0.26
HCR-CS/D&L	35.46 ± 0.07	12.36 ± 0.21	24.72 ± 0.11	28.83 ± 0.17	18.84 ± 0.21	46.03 ± 0.07	11.78 ± 0.33

**Table 6 micromachines-13-01265-t006:** Change in cellular composition (%) of leukocytes. Reproduced from Rapa et al. [116] under a Creative Common CC BY license.

Samples	Time	Neutrophils	Lymphocytes	Eosinophils	Monocytes	Basophils
Control	24 h	28.3 ± 9.5	65.2 ± 19.1	0.1 ± 0.05	6.2 ± 1.3	0.2 ± 0.1
	7 days	28.6 ± 9.3	64.7 ± 18.7	0.1 ± 0.05	6.4 ± 1.1	0.2 ± 0.05
HCB-CS	24 h	27.8 ± 9.7	65.6 ± 18.9	0.1 ± 0.05	6.3 ± 1.1	0.2 ± 0.1
	7 days	28.6 ± 9.5	64.6 ± 19.5	0.2 ± 0.05	6.4 ± 1.5	0.2 ± 0.1
HCB-CS/AGEO	24 h	27.6 ± 9.1	65.9 ± 19.3	0.2 ± 0.1	6.1 ± 1.3	0.2 ± 0.05
	7 days	28.5 ± 9.3	65.0 ± 17.9	0.1 ± 0.05	6.2 ± 1.1	0.2 ± 0.05
HCB-CS/MOEO	24 h	28.3 ± 9.7	65.2 ± 19.1	0.1 ± 0.05	6.2 ± 1.3	0.2 ± 0.05
	7 days	28.8 ± 8.9	64.5 ± 18.5	0.2 ± 0.05	6.3 ± 1.3	0.2 ± 0.1
HCB-CS/AGEO and MOEO	24 h	28.4 ± 8.3	64.9 ± 19.7	0.2 ± 0.05	6.3 ± 1.1	0.2 ± 0.05
	7 days	28.7 ± 8.5	64.5 ± 19.3	0.2 ± 0.05	6.4 ± 1.5	0.2 ± 0.1
HCR-CS	24 h	28.3 ± 9.1	65.1 ± 19.5	0.2 ± 0.1	6.2 ± 1.1	0.2 ± 0.05
	7 days	28.5 ± 8.3	64.8 ± 19.3	0.2 ± 0.05	6.3 ± 1.1	0.2 ± 0.05
HCR-CS/AGEO	24 h	28.6 ± 8.7	65.0 ± 19.1	0.1 ± 0.05	6.1 ± 1.3	0.2 ± 0.05
	7 days	28.5 ± 8.5	64.8 ± 18.7	0.1 ± 0.05	6.4 ± 1.5	0.2 ± 0.05
HCR-CS/MOEO	24 h	27.8 ± 9.3	65.7 ± 18.5	0.1 ± 0.05	6.2 ± 1.3	0.2 ± 0.05
	7 days	28.3 ± 8.9	65.1 ± 19.3	0.2 ± 0.05	6.2 ± 1.3	0.2 ± 0.1
HCR-CS/AGEO and MOEO	24 h	27.6 ± 8.5	65.7 ± 19.5	0.2 ± 0.1	6.3 ± 1.1	0.2 ± 0.1
	7 days	28.7 ± 9.1	64.7 ± 19.1	0.1 ± 0.05	6.3 ± 1.3	0.2 ± 0.05

**Table 7 micromachines-13-01265-t007:** Antibacterial activity of 100% HPMC-based CS–SiNPs-encapsulated PO-PE against *Staphylococcus aureus* (Gram-positive bacteria) and *Escherichia coli* (Gram-negative bacteria) [123].

Storage Time (Days)	Bacterial Inhibition Rate (%)	
	*S. aureus*	*E. coli*
0	98.2 ± 1.25	97.4 ± 1.19
5	99.4 ± 1.26	96.7 ± 1.46
10	97.7 ± 1.13	95.4 ± 1.32
20	96.1 ± 1.04	95.2 ± 1.41
30	92.1 ± 1.21	91.4 ± 1.15
60	89.1 ± 1.32	85.4 ± 1.77

**Table 8 micromachines-13-01265-t008:** Particle size, zeta potential, polydispersity index (PDI), and retention value of clove essential oil-loaded CSNPs [139].

Chitosan/Essential Oil Ratio	Particle Size (nm)	Zeta Potential (mV)	PDI	Retention of Essential Oil (%)
1:0	223.2 ± 35.6	+34.50 ± 1.6	0.337 ± 0.018	-
1:0.25	265.1 ± 18.2	+20.14 ± 0.7	0.264 ± 0.013	55.8 ± 3.9
1:0.50	295.8 ± 45.6	+16.50 ± 1.6	0.221 ± 0.005	73.4 ± 4.8
1:1	444.5 ± 63.6	+10.14 ± 0.7	0.117 ± 0.025	63.1 ± 5.5

**Table 9 micromachines-13-01265-t009:** Antibacterial evaluation of CSNP nanoencapsulation of lemongrass essential oil.

Bacteria	Microbial Method	Lemon Grass Essential Oil	CSNPs	Essential Oil Loaded CSNPS	Control Drugs
					Streptomycin
*E. coli*	MIC (%)	3.12	25	6.25	0.19
	MMC (%)	6.25	50	12.5	0.39
	DIZ (mm)	22.3	2.5	13.8	19.5
*L. monocytogenes*	MIC (%)	3.12	25	6.25	0.09
	MMC (%)	3.12	25	6.25	0.19
	DIZ (mm)	24.5	3.2	15.5	21.3
*S. aureus*	MIC (%)	0.39	12.5	1.56	0.05
	MMC (%)	0.78	25	3.12	0.10
	DIZ (mm)	27.8	6.5	17.5	24.5

## Data Availability

Not applicable.

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
