# Peer review of "Chitosan Nanoparticle Encapsulation of Antibacterial Essential Oils"

_micromachines, 2022, doi:10.3390/mi13081265_

Round 1

Reviewer 1 Report

1.     There are many varieties of essential oils available. Why author selected only BEO, MEO, CEO, and CA. Moreover, the author needs to add a few more essential oils such as  Anethum graveolens Essential Oil, Encapsulation of Cardamom Essential Oil, Peppermint oil (PO), and Green Tea oil (GTO). These oils are also encapsulated with Chitosan nanoparticles. Why author didn’t include these essential oils in this review.

2.     Suggestions: This review mainly focuses on Chitosan and essential oils. Therefore encapsulation of essential oils should be focused more than on why chitosan was selected among other polysaccharides. However, as given in comment 1, adding a few more essential oil (if possible), may improve the quality of the manuscript.   

3.     Figure 2 is not mentioned anywhere in the text.

4.     Page 4, Line 78, “S. aureus biofilm formation in-vivo orthopedic implant Figure 3”. Rewrite the sentence or put figure 3 in the bracket

5.     Like pectin and dextran, chitosan is also a polysaccharide. Why author separates the chitosan as a new section, not given under section 3 (as 3.5).

6.     The abbreviations for TPP and HMP should be given where it appears first in the manuscript.

7.     Section 5.2 first line “Wu and Liu coworkers” reference number need to be cited since this reference couldn’t find. Moreover, Ref 55 and 57 are given only at the Figure/table caption, not in the text.

8.     “Nanoencapsulation of Cinnaldehyde (CA)” should be section 5.4 (not 5.3). Cinnaldehyde should be Cinnamaldehyde.

Author Response

We thank the reviewer for taking the time and considering the paper. We address the comments below.

The changes are marked in Yellow color in the manuscript.

  1. There are many varieties of essential oils available. Why author selected only BEO, MEO, CEO, and CA. Moreover, the author needs to add a few more essential oils such as  Anethum graveolens Essential Oil, Encapsulation of Cardamom Essential Oil, Peppermint oil (PO), and Green Tea oil (GTO). These oils are also encapsulated with Chitosan nanoparticles. Why author didn't include these essential oils in this review.
  2. Suggestions: This review mainly focuses on Chitosan and essential oils. Therefore encapsulation of essential oils should be focused more than on why Chitosan was selected among other polysaccharides. However, as given in comment 1, adding a few more essential oil (if possible), may improve the quality of the manuscript.   

Reply to comments 1 and 2:

Required information regarding the essential oil is added

To the previous essential oils: Basil essential oil (Ocimum basilicum), Mandarin essential oil (Citrus reticulata), Carum copticum essential oil ("Ajwain"), Cinnamaldehyde

Following essential oils are now added.

Dill plant seed essential oil (Anethum graveolens), Peppermint oil (Mentha piperita) and Green tea oil and (Camellia sinensis), Cardamom essential oil, Clove essential oil (Eugenia caryophyllata), Cumin seeds essential oil (Cuminum cyminum), Lemongrass essential oil (Cymbopogon commutatus), Summer savory essential oil (Satureja hortensis), Thyme essential oil, Cinnamomum essential oil (Cinnamomum zeylanicum), Nettle essential oil (Urtica dioica). Major essential oil components carvacrol; oil-in-water nanoemulsion of Eucalyptus oil (Eucalyptus globu-lus), Mandarin essential oil, Electrospinning nanofiber of collagen hydrolysate-chitosan with lemon balm (Melissa officinalis) and dill (Anethum graveolens) essential oil.

  1. Figure 2 is not mentioned anywhere in the text.

Figure 2 description of the running text is now added on Page 2, lines 73 and 74

  1. Page 4, Line 78, "S. aureus biofilm formation in-vivo orthopedic implant Figure 3". Rewrite the sentence or put figure 3 in the bracket.

The sentence was incorrect and didn't match the paragraph's context; therefore, this line is removed in the revised manuscript.

  1. Like pectin and dextran, Chitosan is also a polysaccharide. Why author separates the Chitosan as a new section, not given under section 3 (as 3.5).

We understand the concern of the reviewer.

However, we think the flow of the paper and the reader's engagement on the topic "chitosan-based encapsulation of essential oil" would be better if we separate this section as Section 4. Also, this would further improve the readability by showcasing the significance of "Chitosan as polymeric material" compared to the other polysaccharides mentioned in Section 3.

  1. The abbreviations for TPP and HMP should be given where it appears first in the manuscript.

Provided on page 10, lines 287 and 288

  1. Section 5.2 first line "Wu and Liu coworkers" reference number need to be cited since this reference couldn't find. Moreover, Ref 55 and 57 are given only at the Figure/table caption, not in the text.

 Page 11, line 345 reference is highlighted in yellow color.

  1. Nanoencapsulation of Cinnaldehyde (CA)" should be section 5.4 (not 5.3). Cinnaldehyde should be Cinnamaldehyde.

Corrected as Page 17 line 506

Reviewer 2 Report

Dear Authors

The presented review is very interesting and addresses a very important issue. A very important problem has been discussed and a declaration of the successive solutions has been presented.

However, some essential points have been missed out such as:

1- The development of Chitosan derivatives and their use in biofilm resistance.

2- The quaternization techniques used in the chitosan quaternization process.

3- The techniques used in the preparation of chitosan nanoparticles.

All the abovementioned comments should be addressed clearly before your review can be considered for publication. 

Author Response

We thank the reviewer for taking the time and considering the paper. We address the comments below.

The changes are marked in Cyan color in the manuscript.

  • The development of Chitosan derivatives and their use in biofilm resistance.

2- The quaternization techniques used in the chitosan quaternization process.

Reply to Comment 1 and 2

Page 9 line 235, a new section is added “4.1 Different forms of Chitosan and its quaternization as antibiofilm activity”

3- The techniques used in the preparation of chitosan nanoparticles.

To compile all reported the methods for chitosan nanoparticles preparation is beyond the scope of this paper. Although, we understand the concern of the reviewer and therefore we summaries the methods and a new table is dedicated as Table 1. Chitosan nanoparticles preparation methods

Reviewer 3 Report

Thorough review and well presented! Could you please add caption descriptions for Figures 2B, 2C and 2D?

Author Response

We thank the reviewer for taking the time and considering the paper. 

The Figure captions are now added for 2b, 2c, 2d

Round 2

Reviewer 2 Report

Dear Authors

Thank you very much for taking my comments into your consideration during the revision process.

I can recommend the current revised version of your manuscript for publication.